# A Combined Statistical Bias Correction and Stochastic Downscaling Method for Precipitation

Claudia Volosciuk[1], Douglas Maraun[2], Mathieu Vrac[3], and Martin Widmann[4]

[1]GEOMAR Helmholtz Centre for Ocean Research Kiel, Kiel, Germany.
[2]Wegener Center for Climate and Global Change, University of Graz, Austria.
[3]Laboratoire des Sciences du Climat et de l'Environnement (LSCE), IPSL, Gif-sur-Yvette, France.
[4]School of Geography, Earth and Environmental Sciences, University of Birmingham, United Kingdom.

*Correspondence to:* Claudia Volosciuk (cvolosciuk@geomar.de)

**Abstract.** Much of our knowledge about future changes in precipitation relies on global (GCM) and/or regional climate models (RCM) that have resolutions which are much coarser than typical spatial scales of precipitation, particularly extremes. The major problems with these projections are both climate model biases and the gap between gridbox and point scale. Wong et al. developed a model to jointly bias correct and downscale precipitation at daily scales. This approach, however, relied on pairwise correspondence between predictor and predictand for calibration, and thus, on nudged simulations which are rarely available. Here we present an extension of this approach that separates the downscaling from the bias correction and in principle is applicable to free running GCMs/RCMs. In a first step, we bias correct RCM-simulated precipitation against gridded observations at the same scale using a parametric quantile mapping ($QM_{grid}$) approach. In a second step, we bridge the scale gap: we predict local variance employing a regression based model with coarse-scale precipitation as predictor. The regression model is calibrated between gridded and point scale (station) observations. For this concept we present one specific implementation although the optimal model may differ for each studied location. To correct the whole distribution including extreme tails we apply a mixture distribution of a gamma distribution for the precipitation mass and a generalized Pareto distribution for the extreme tail in the first step. For the second step a vector generalized linear gamma model is employed. For evaluation we adopt the perfect predictor experimental setup of VALUE. We compare our method also to the classical QM as it is usually applied, i.e., between RCM and point scale ($QM_{point}$). Precipitation is in most cases improved by (parts of) our method across different European climates. The method generally performs better in summer than in winter and in winter best in the Mediterranean region with a mild winter climate and worst for continental winter climate in mid & eastern Europe or Scandinavia. While $QM_{point}$ performs similar (better for continental winter) to our combined method in reducing the bias and representing heavy precipitation it is not capable to correctly model point scale spatial dependence of summer precipitation. A strength of this two-step method is that the best combination of bias correction and downscaling methods can be selected. This implies that the concept can be extended to a wide range of method combinations.

# 1 Introduction

To assess the impacts of hydrometeorological extremes in a changing climate high quality precipitation projections on the point scale are often demanded. Much of our knowledge about future changes in precipitation is based on global (GCMs) and/or regional climate models (RCMs). These have resolutions which are much coarser than typical spatial scales of processes relevant for precipitation. This concerns particularly extreme precipitation, which is far more sensitive to resolution than mean precipitation (Volosciuk et al., 2015). Although horizontal resolution of GCMs has successively increased since the first assessment report of the Intergovernmental Panel on Climate Change (IPCC, 1990), resolving all important spatial and temporal scales remains beyond current computational capabilities for transient global climate change simulations (Le Treut et al., 2007). The simulation of precipitation depends heavily on processes that are parameterized in current GCMs, and also in most RCMs (Flato et al., 2013). Biases related to parametrization schemes and unresolved processes thus remain in addition to systematic biases related to the large scale circulation (e.g., Flato et al., 2013; Kotlarski et al., 2014).

Different approaches have been employed to downscale and/or reduce biases of simulated precipitation, particularly extremes: (a) high-resolution GCMs, (b) dynamical downscaling using RCMs that are nested into the GCMs (Rummukainen, 2010), and (c) statistical downscaling including post-processing with bias correction methods (Maraun et al., 2010). But even though high-resolution GCMs and RCMs improve the representation of extreme precipitation by better resolving mesoscale atmospheric processes, biases remain and there is still a scale gap between the simulated gridbox values of precipitation and point scale data (i.e., rain gauges). Hence, statistical bias correction methods are also applied to such high-resolution simulations. These so-called Model Output Statistics (MOS)-approaches employ a correction function derived in present day simulations to future simulations of the same model (Maraun et al., 2010).

Quantile mapping (Piani et al., 2009), one example MOS-approach, is widely applied to statistically post-process simulated precipitation. While this might be a reasonable approach for correcting biases on the same spatial scale, variability on local scales is not fully determined by grid-scale variability, e.g., the exact location, size or intensity of a thunderstorm. This is part of the representativeness problem between gridbox and point values (Zwiers et al., 2013). Quantile mapping is a deterministic approach that cannot add random variability. It simply inflates the variance leading to an overestimation of spatial extremes, and too smooth a variance in space and also in time (von Storch, 1999; Maraun, 2013a). Grid-box precipitation, e.g., is the area average of sub-grid precipitation. The aggregation averages local variations in time such that grid-box time series are smoother in time than local time series. Quantile mapping can not overcome this mismatch in temporal structure (apart from correcting the drizzle effect). This temporal effect is more difficult to trace than the spatial effect (Maraun, 2013a). Standard downscaling approaches in turn have a limited ability to correct systematic biases. Wong et al. (2014) developed a model that jointly bias corrects and downscales precipitation at daily scales. However, this approach relies on pairwise correspondence between predictor and predictand for calibration that is only provided by nudged GCM/RCM-simulations, and is not able to post-process standard, free-running GCM-simulations (Eden et al., 2014).

Here we present a modification of the Wong et al. (2014)–approach that is designed to also work in principle for free running GCM/RCMs, such as those available from ENSEMBLES (van der Linden and Mitchell, 2009) or CORDEX (e.g.,

Jacob et al., 2013). With the aim of combining their respective advantages we combine a statistical bias correction and a stochastic downscaling method. Thereby we separate bias correction from downscaling by inserting a gridded observational dataset as reference between these two steps. In particular, as first step we apply a parametric quantile mapping approach between an RCM and a gridded observational dataset. In a second step we bridge the scale gap between gridded and point scale by employing a stochastic regression-based model that is calibrated between gridded and station observations and then applied to the bias corrected precipitation from the first step.

In section 2 the general concept is introduced, the data used are described in section 3. In section 4 we present the bias correction and the stochastic downscaling model. Results of the evaluation of our model for example stations across Europe are provided in section 5 and finally, section 6 contains the conclusion.

## 2  General concept

We separate bias correction from downscaling into two steps to overcome the shortcomings of each method and to combine their respective strengths. Our concept is illustrated schematically in Fig. 1. In the first step, we use the advantage of distribution-wise bias correction (i.e., the correction function is calibrated on long-term distributions) to eliminate systematic biases in the RCM. While this distribution-wise setting may correct systematic RCM-biases it cannot bridge the gap between gridbox and point scale for two reasons. First, a considerable portion of subgrid variability is random for precipitation and has to be modeled as stochastic noise. Yet, distribution-wise MOS-methods are deterministic and do thus not add unexplained random variability. Second, distribution-wise methods cannot separate local variability into systematic explained variability and small-scale unexplained variability. Moreover, when simulated short-term variability is inflated to match local variability, long-term trends are also inflated (Maraun, 2013a). Therefore, we only apply this distribution-wise method to correct biases on the same spatial scale, i.e., as reference we use gridded observations on the same grid as the RCM. In the second step, we employ a stochastic regression-based model to overcome the representativeness problem. This regression model corrects systematic local effects (e.g., whether a rain gauge is positioned on the lee or windward side of a mountain). It also adds random (unexplained) small scale variability, in contrast to approaches of combined methods that employ spatial interpolation for downscaling (Wood and Maurer, 2002; Wood et al., 2004; Payne et al., 2004) or rescale the grid-scale precipitation with a factor to match the observations (Ahmed et al., 2013). We calibrate the probabilistic regression model between gridded and point scale observations and apply it then to the corrected grid-scale time series in the validation period. This corresponds to a perfect prog (PP) setting for the regression, while the bias is corrected in the first step. This combined approach is an extension of the model by Wong et al. (2014) that jointly bias corrects and downscales precipitation (see Fig. 1). They employ a probabilistic regression model that is calibrated between RCM and point scale observations (MOS-approach). It requires nudged RCM-simulations for calibration since temporal correspondence is essential.

[Figure 1 about here.]

With this concept in place, basically in the first step any reasonable distribution-wise MOS-approach, and in the second step any adequate stochastic model can be employed. A strength of this concept is its flexibility, i.e., the best suitable combination

of statistical models for a given location and season can be determined. In this study, we employ a quantile mapping ($QM_{grid}$) approach based on the mixture distribution of a gamma and a generalized Pareto distribution (Vrac and Naveau, 2007) in the first step. The model used in the second step consists of a logistic regression for wet day probabilities and a vector generalized linear model predicting the parameters of a gamma probability distribution (VGLM gamma) for precipitation intensities. Note that this combination of methods may not be optimal in all studied locations. However, the aim of this study is rather to introduce and evaluate the concept of this combined approach than to find the optimal specific implementation for all studied locations.

To evaluate and illustrate our method we adopt the perfect predictor experimental setup of the VALUE-framework (Maraun et al., 2015). Employing the same evaluation framework as VALUE allows for comparing our method to all models participating in the VALUE-experiment. In this context, a reanalysis-driven RCM is used which allows to evaluate the ability of the method to correct RCM-biases, before evaluating GCM-driven simulations where biases of both GCM and RCM need to be corrected. We note that although this is a pairwise setup where simulated and observed weather states are in principle synchronized (with the exception of the internal variability generated within the RCM) we only use the simulated and observed distribution for the bias correction. Thus, as explained above, the approach can be transferred to any simulation setup, e.g., GCM-driven RCM-simulations or GCM-simulations. For comparison we also applied the classical QM-approach, i.e., directly between RCM and point scale ($QM_{point}$).

The method is evaluated by five-fold cross validation for the time period 1979-2008, i.e., five 6-year long periods are predicted by the model that was fitted to the remaining 24 years. Artificial predictive skill is thus not present as the predicted period is not part of the training period. The model is fitted and evaluated for each season separately. 86 stations across Europe are studied (as selected for the VALUE experiment, see Fig. 2) representing different climates. In the evaluation of our model we compare eight European subdomains (dashed lines in Fig. 2): the British Isles (BI), the Iberian Peninsula (IP), France (FR), Mid-Europe (ME), Scandinavia (SC), the Alps (AL), the Mediterranean (MD) and eastern Europe (EA). These domains have been defined within the PRUDENCE-project (Christensen and Christensen, 2007) and are often used for RCM-evaluation (e.g., Kotlarski et al., 2014). Although climatic differences within these subdomains remain they summarize European climate zones and intercomparison amongst them allows for studying large-scale gradients (e.g., from maritime (west) to continental (east) or from cold (north) to mild (south) winters). We slightly extended the PRUDENCE-regions SC, AL and MD such that all studied rain gauges are included in the analysis.

[Figure 2 about here.]

## 3  Data and grid box selection

As prescribed by the perfect predictor experiment within the VALUE framework we use the RCM RACMO2 from the KNMI (van Meijgaard et al., 2012) to test our method for the time period from 1979–2008. The RCM has been driven with ERA-Interim reanalysis (Dee et al., 2011) within the EURO-CORDEX framework (Jacob et al., 2013). The simulation has been

carried out at a horizontal resolution of 0.44° (∼50 km) over a rotated grid. Note that the resolution we employ (0.44°) differs from the resolution used in the VALUE experiment (0.11°).

As gridded observational dataset E-OBS version 10 (Haylock et al., 2008) is used, also at 0.44° resolution. The reason for chosing the 0.44° horizontal resolution for both RCM and E-OBS is that the actual resolution of E-OBS might in some regions be lower than the nominal 0.22° due to sparse rain gauge density included in the dataset[1]. Gridding very few rain gauges to a high resolution might in particular result in too smooth extremes (Haylock et al., 2008; Hofstra et al., 2009a, b; Maraun et al., 2011a). Hence, too high a resolution of a gridded dataset may be an unreliable reference for bias correction, at least for summer extreme events. Moreover, this could cause artificial smoothing of extremes by bias correction. In some regions where station density is very sparse this might even hold true for the chosen resolution. Although E-OBS is probably not an appropriate reference in some regions it is the best available gridded dataset covering the whole EURO-CORDEX domain.

The E-OBS reference gridbox for both steps (bias correction and downscaling) is generally the closest gridbox to the respective station. If the closest gridbox is an ocean gridbox (i.e., for coastal and island stations) and only contains missing values we select the gridbox with the highest correlation in winter between daily precipitation at the given station and the five closest E-OBS-gridboxes. In winter the spatial decorrelation length of precipitation is generally large implying that often several gridboxes are affected by the same weather system, and thus, the gridbox with the most similar climate can be reliably identified.

The RCM-gridbox that is bias-corrected and downscaled is generally chosen as the closest gridbox to the E-OBS reference gridbox – also for coastal and island stations where the chosen RCM-gridbox might thus differ from the closest RCM-gridbox to the final reference (i.e., rain gauge). For locations in the rain shadows we choose the RCM-gridbox which best represents the climate at the given location to correct too low precipitation values caused by not enough windward air masses crossing the mountain range ("location bias", Maraun and Widmann, 2015). To this end, the highest correlation between the winter seasonal mean of RCM and gridded observations within 250 km around the closest gridbox to the observations is determined. Note that when transferring this approach to free running RCM-simulations this gridbox selection step needs to be carried out employing a reanalysis-driven simulation of the same RCM to ensure temporal correspondence.

For local scale observations we used 86 stations across Europe from ECA&D (Klein Tank et al., 2002) selected by the VALUE experimental framework (Maraun et al., 2015). The locations and ids of these stations are illustrated in Fig. 2. A detailed analysis is carried out for some example stations representing different climates (highlighted in blue in Fig. 2).

## 4 Statistical Model

### 4.1 Step 1: Bias correction

In our model we correct several biases. In a first step, the "location bias" is corrected by gridbox selection (see section 3 for details). In the second step, the "drizzle" effect is corrected by increasing the wet day threshold for the RCM such that

---

[1]For station density of actual E-OBS versions refer to the ECA&D website: http://www.ecad.eu/dailydata/datadictionary.php

the number of wet days (closely) matches the gridded observations with a threshold of $0.1$ mm d$^{-1}$. Finally, we correct precipitation intensities of wet days (i.e., exceeding the corrected wet day threshold) using a quantile mapping (QM) approach which is described in the following. The correction function $y = f(x)$ between the simulated ($x$) and the corrected ($y$) values of daily precipitation intensities such that the corrected values match the observations is based on the cumulative distribution functions (cdfs) as: $\text{cdf}_{obs}(f(x)) = \text{cdf}_{RCM}(x)$ (Piani et al., 2009). To allow for extrapolation in a future climate to unobserved precipitation intensities and to avoid deterioration of future extremes that might occur with an approach that relies on empirical cdfs we chose a parametric QM-approach.

To model precipitation intensities the gamma distribution is commonly used (Katz, 1977). While the bulk of precipitation is generally well represented the tail of the gamma distribution is usually too light to capture high and extreme rainfall intensities (e.g., Vrac and Naveau, 2007; Maraun et al., 2010). Thus, an extreme value distribution, such as the generalized Pareto (GP) distribution (Coles, 2001), might be required to model the extremes of the precipitation distribution. To correct the .whole precipitation distribution including extreme tails we apply the mixture distribution of Vrac and Naveau (2007) which consists of a gamma distribution for the precipitation mass and a GP distribution for the extreme tail. This model is a variant of Frigessi et al. (2002). The distribution $l_\phi(x)$ of observed precipitation $x$ on wet days is modeled as

$$l_\phi(x) = c(\phi)\left(\left\{[1 - w_{m,\tau}(x)]\, f_{\lambda,\gamma}(x)\right\} + [w_{m,\tau}(x) g_{\xi,\sigma}(x)]\right),$$
$$\phi = (\lambda, \gamma, \xi, \sigma, m, \tau), \tag{1}$$

where $f_{\lambda,\gamma}$ is the probability density function (pdf) of the gamma distribution with the rate parameter $\lambda$ and the shape parameter $\gamma$,

$$f_{\lambda,\gamma}(x) = \frac{\lambda^\gamma}{\Gamma(\lambda)} x^{\gamma-1} e^{-\lambda x}, \qquad \lambda, \gamma > 0, \tag{2}$$

and $g_{\xi,\sigma}$ is the pdf of the GP distribution:

$$g_{\xi,\sigma}(x) = \frac{1}{\sigma}\left[1 + \frac{\xi(x-u)}{\sigma}\right]^{-\frac{1}{\xi}-1} \qquad \text{for} \quad x \geq u, \tag{3}$$

with the scale parameter $\sigma > 0$ and the shape parameter $\xi$ which determines the tail behavior of the GP distribution as follows: $\xi < 0$: bounded tail; $\xi \to 0$: exponential distribution (light tailed); and $\xi > 0$: infinite heavy tail. Here, we constrain $\xi \geq 0$ to ensure that our model can be applied to a future climate that may experience higher values than those observed during the present day training period for the model. The function $w_{m,\tau}$ is a weight function that determines the transition between the gamma and GP pdfs as

$$w_{m,\tau}(x) = \frac{1}{2} + \frac{1}{\pi} \arctan\left(\frac{x-m}{\tau}\right), \qquad m, \tau > 0, \tag{4}$$

with the location parameter $m$ denoting the location of the center of this transition and the transition rate $\tau$ influencing the rapidity of the transition between the two distributions. To finally obtain the mixture pdf the mixture function (Eq. 1) must be normalized which is carried out here by multiplying the mixture function by a constant $c(\phi)$. In the mixture pdf (Eq. 1) the threshold $u$ in the GP distribution (Eq. 3) is set to zero, as the location parameter $m$ of the weight function (Eq. 4) fulfills the purpose of a threshold in Eq. 1. Moreover, setting the threshold to zero and applying a weight function instead solves also the problem of threshold selection with unsupervised estimation and avoids discontinuity in the mixture pdf $l_\phi(x)$ (Eq. 1) (Vrac and Naveau, 2007). The parameters for $l_\phi(x)$ are estimated using maximum likelihood estimation (MLE). For technical details on the implementation of this model please refer to appendix A1.

Since the mixture model is a complex model with six free parameters, a thorough statistical model selection is necessary. We select between the mixture model and the simpler gamma only model separately for the observed ($F_{obs}$) and RCM-simulated ($F_{RCM}$) distributions. For the selection, we apply the Akaike information criterion (AIC, Akaike, 1973), which asymptotically selects the model that minimizes the mean squared error between prediction and observation (Shao, 1997). The AIC is defined as $-2\log(L) + 2k$ with the likelihood $L$ corresponding to the maximum likelihood estimate of the $k$ model parameters. The AIC is dominated by the most densely populated region of the distribution. Hence, a good fit for the bulk of the distribution (and thus a low AIC) might nevertheless come along with large biases in the extremes (see appendix A2 for an example). To avoid a model choice with unreasonably high extremes we therefore introduce a criterion based on a comparison between the 100 season return levels estimated by the mixture model (Eq. 1) and by the GP distribution (Eq. 3) before the AIC-based model selection is applied. For technical details on these model selection procedures please refer to appendix A2.

To strictly avoid that bias correction deteriorates the predictor and introduces biases both the complete cross-validated corrected time series and the raw RCM output are compared to gridded observations as reference using the Cramér-von Mises (CvM) criterion. The CvM is a measure for the distance between two empirical cdfs (cdf-bias hereafter; Darling, 1957) and has been used to evaluate cdf-based correction models before (e.g., Michelangeli et al., 2009; Vrac et al., 2012). If $\text{cdf}_{ref}(x)$ is the empirical cdf of observations as reference (i.e., the perfect bias correction would match this reference) and $\text{cdf}_{corr}(x)$ is the empirical cdf of the bias corrected time series the CvM-statistics is defined as the integrated squared difference between $\text{cdf}_{ref}$ and $\text{cdf}_{corr}$ as follows

$$\text{CvM} = \int\limits_{-\infty}^{\infty} | \text{cdf}_{corr}(x) - \text{cdf}_{ref}(x) |^2 \, dx \tag{5}$$

Here, the CvM is computed for both the corrected daily precipitation time series and the uncorrected RCM simulated precipitation time series with E-OBS as reference. The predictor for downscaling is selected based on the lower CvM. In other words, the bias corrected time series is only used as predictor for the downscaling step if it improves the predictor compared to the raw uncorrected RCM.

## 4.2 Step 2: Stochastic downscaling

To bridge the scale gap we apply the regression model developed by Wong et al. (2014) as follows. We determine the statistical relationship between gridded and station observations. This statistical relationship is then applied to coarse-scale precipitation as predictor which is selected in the first step, i.e., $\text{QM}_{grid}$-bias corrected or uncorrected RCM-simulated precipitation. To be able to estimate the distribution of precipitation as a function of a given predictor a stationary distribution is not sufficient. The family of generalized linear models (GLMs) extends linear regression to such purposes (e.g., Dobson, 2001). In this framework the time-dependent expectation of a random variable is linked via a monotonic link function to a linear combination of predictors. The logistic regression model belongs to the class of GLMs and is often used to model the changing probability of rainfall occurrence (Chandler and Wheater, 2002). We model the probability $p_i$ of a day $i$ being wet (i.e., greater than the threshold selected earlier at 0.1 mm d$^{-1}$) as a function of coarse-scale precipitation $x_i$ as

$$h(p_i) = \log\left(\frac{p_i}{1-p_i}\right) = \alpha x_i + \beta \tag{6}$$

where $h(\cdot)$ is the logit link function and the parameters $\alpha$ and $\beta$ are estimated by MLE. The logit link function gives the logarithm of the odds.

Subsequently, precipitation intensity on wet days is modeled using a vector generalized linear model (VGLM) as regression model (Yee and Wild, 1996; Yee and Stephenson, 2007). VGLMs are an extension of GLMs. While GLMs describe the conditional mean of a wide range of distributions VGLMs allow for prediction of a vector of parameters from the same set of predictors which is useful if one is also interested in the variance or the extremes of a distribution. Wong et al. (2014) implemented a mixture model-version (see Eq. 1) and a gamma model-version (see Eq. 2) employing a VGLM. Here we apply the VGLM gamma version since the calibration and model selection procedure for the VGLM mixture model is computationally rather expensive. The simpler gamma model might be sufficient here as in the downscaling step a predictor is employed that already explains a large portion of the variance. The quality of downscaled precipitation does not only depend on the chosen model but also on the quality of the predictor. Employing the mixture model for the bias correction step is thus meaningful to ensure a good representation of higher quantiles and extremes in the predictor although downscaling is performed with a simpler gamma model. The scale $\theta$ (the inverse of $\lambda$ in Eq. 2) and the shape $\gamma$ parameters depend linearly on the predictor (coarse-scale precipitation) $x_i$. The model has the form

$$\theta_i = \theta_0 + \psi_\theta x_i$$
$$\gamma_i = \gamma_0 + \psi_\gamma x_i \tag{7}$$

where the regression parameters $\psi_\theta$, and $\psi_\gamma$ are estimated by MLE.

Combining the probability of wet day occurrence and the gamma model distribution defining the precipitation intensities we get the probability that observed precipitation on a given day ($R_i$) is less than or equal to a particular precipitation intensity ($r$):

$$\Pr_{\theta,\gamma}(R_i \leq r) = \Gamma_{\theta,\gamma}(R_i \leq r \mid W)p_i + (1 - p_i) \tag{8}$$

where $\Gamma_{\theta,\gamma}(R_i \leq r \mid W)$ is the gamma-cdf and $p_i$ is the probability of that given day being wet.

## 4.3 Evaluation metrics

We evaluate our combined model based on the following metrics:

- *Mean bias*: Absolute difference between seasonal means as $(\mathrm{model} - \mathrm{reference})$.

- *cdf-bias*: Cramér-von Mises (CvM) criterion which represents the mean squared error of a cdf compared to a reference cdf (for details see section 4.1).

- *%sim > perc95*$_{\mathrm{obs}}$: percentage of simulated wet days exceeding the observed 95th percentile.

- *QQ-Plots*: The quantiles (i.e., sorted time series) of modeled precipitation are plotted against the quantiles of the reference. For the evaluation of the second step (downscaling) standardized QQ-plots are used which are explained in section 5.2.2.

- *Spatial autocorrelation*: Correlation of a variable with itself in geographical space. The correlogram is estimated by centred Mantel statistic using the R-package ncf (Bjornstad, 2015). The correlation for a set of distances at discrete distance classes is calculated. Significance is assessed by 1000 random permutations. The correlogram is estimated for daily values and then averaged. For the VGLM the correlogram is computed for 100 realisations of the stochastic model and then averaged. The correlogram is centred on zero, i.e., zero represents similarity across the region. Crossing the zero-line implies thus that the pair of distances is not more similar than what would be expected by chance alone across the region.

## 5 Results

We first evaluate the mean bias of our combined model (selected predictor & VGLM) against station observations and compare it to the raw uncorrected RCM and to classical QM$_{point}$ (between RCM and point scale). Then the performance of the two steps (bias correction and downscaling) is assessed individually and in combination. Finally, all analysed models are compared. The evaluation is carried out for the time period 1979–2008 by analyzing the cross-validated (five-fold) time series'. The first step (bias correction) is evaluated against the gridded E-OBS dataset although E-OBS might underrepresent the extremes in some regions where station density is sparse. The second step (downscaling) and the combined model (step 1 & 2) are evaluated against station observations.

## 5.1 Evaluation of mean precipitation bias

Figure 3 shows the mean bias of precipitation (against station observations) as modeled by (a, b) the RCM , (c, d) the classical $QM_{point}$-approach applied directly between RCM and station observations and (e, f) our combined model. The RCM has a stronger bias in DJF than in JJA. In DJF it is rather too wet whereas in JJA many locations have a dry bias. In both seasons the bias is improved by $QM_{point}$ with a slight remaining wet bias. Our combined model also improves the mean bias of the RCM in JJA. Yet, in DJF wet biases remain and got even worse in some locations. This raises the question why the results become worse when statistical post-processing is applied. Yet, the bias of the seasonal mean does not give information on how the precipitation distribution is represented nor the predictive power of the model. These issues are evaluated in the following.

[Figure 3 about here.]

## 5.2 Evaluation of Combined Model

First both steps of the combined model are evaluated individually. Second, the combination of both steps is evaluated. In this combined model the predictor selected in the first step is used for the regression model in the second step.

### 5.2.1 Evaluation of Step 1: Bias Correction vs. E-OBS

Figure 4 shows the cross-validated selected predictor (uncorrected RCM: triangles, $QM_{grid}$-corrected RCM: circles) that is used in the second step for downscaling. For predictor selection we apply the Cramér-von Mises-score (CvM, Eq. 5, section 4.1) which represents the mean squared error of a cdf compared to a reference cdf (cdf-bias hereafter). The predictor is selected based on the lowest CvM-score of the cross-validated $QM_{grid}$-corrected time series and the raw uncorrected RCM with gridded observations as reference. Generally our bias correction often improves precipitation. It is selected 73 times in December–February (DJF) and 49 times in June–August (JJA) out of 86 rain gauges.

The CvM-values of the selected predictor (Fig. 4a, b) indicate that the cdf-bias is generally lower in JJA than in DJF. In DJF the cdf-bias is lowest in the Mediterranean region with a mild winter climate. Yet, the CvM-criterion is quite sensitive to small deviations between the cdfs. The highest selected CvM-values are found for Graz (Austria) in JJA, and Leba (northern Poland), Siedlce (eastern Poland) and Dresden (eastern Germany) in DJF. QQ-Plots for these high CvM-values (see appendix B1) suggest that the corrected time series are still usable and show improvements compared to the raw RCM although they are of course not a perfect match of the observations. These remaining inaccuracies of the $QM_{grid}$-approach can be related to both a time-varying correction function and the parametric correction function. Figure 5 summarizes Fig. 4 over the European subdomains by boxplots. Spatial variability throughout the subdomain is quantified by CvM-variability represented by the box. In DJF the boxplots confirm the lowest cdf-bias in the Mediterranean region (MD & IP) that is already visible in the map (Fig. 4a). The highest median is in ME. Yet, although the median is slightly lower than in ME spatial variability is largest in EA, extending to the highest CvM-values. This indicates that there are problems with continental winter climate which persist after bias correction as in ME and EA mostly the bias corrected model is selected (Fig. 4a). QQ-Plots of the two worst examples in EA (Leba and Siedlce; Fig. 15) show that the complete precipitation time series remains too wet whereas in the worst example

of ME (Dresden; Fig. 15) the bias correction performs well for most values and only fails in the highest quantile. In JJA the CvM-score, and hence the cdf-bias, is very low and no pronounced differences between the subdomains can be identified (Fig. 5a).

[Figure 4 about here.]

[Figure 5 about here.]

The representation of heavy precipitation by the selected predictor is evaluated by the percentage of simulated values that are higher than the 95th percentile of the observations on wet days (%sim > perc95$_{obs}$, Figs. 4c, d, 5c, d). Thus, in a "perfect" model this would be exactly 5 % (yellow). In many locations there are slightly too many "extremes", i.e., the occurrence of heavy precipitation (> perc95$_{obs}$) is overestimated, particularly in DJF. Consistent with the CvM-score the overestimation in

heavy-precipitation-occurrence increases in DJF from west to east (FR → ME → EA) and is again highest in EA, followed by ME and SC (Fig. 5c). In JJA the occurrence of heavy precipitation is quite well represented in AL and BI (Fig. 5d), it is, however, in some locations underestimated (Fig. 4d). In the other subregions the occurrence of heavy precipitation is also slightly overestimated in JJA (Fig. 5d).

### 5.2.2 Evaluation of Step 2: Downscaling vs. Station

Here we present some examples to illustrate the performance of the VGLM gamma for different climates , calibrated between gridded (E-OBS) and point scale (station) observations. All results that are shown for the evaluation of the downscaling step (step 2, Figs. 6–7 and appendix B2) are calibrated over the complete time period and then predicted by E-OBS as predictor for the same time period. As we do not use the cross-validated time series here the best possible relationship is presented. This allows to evaluate the goodness-of-fit and is a necessary step before evaluating the model in a cross-validation setup. For a

detailed evaluation of the VGLM gamma for the relationship between nudged RCM/GCM-simulations and station observations over the British Isles refer to Wong et al. (2014) and Eden et al. (2014).

To evaluate the goodness-of-fit we use residual QQ-plots (Fig. 6 for DJF and appendix B2 for JJA). As a QQ-plot requires quantiles of an unconditional distribution we standardized the from day-to-day varying distribution to a stationary gamma distribution[2] (Coles, 2001; Wong et al., 2014). This stationary distribution has no longer the predictor-dependent day-to-day

variations, i.e., the effect of the predictor is approximately removed. Due to this procedure the goodness-of-fit of the regression model can be evaluated separately, instead of evaluating only the combined effect of predictor and regression model which is present in the time-varying gamma-parameters, and thus, also in realisations drawn from these varying distributions. Therefore, deficiencies that are indicated by these standardized QQ-plots are either due to inappropriate model structure or not well fitting

---

[2]standardization is performed as: 1) compute probabilities for reference values (here: station observations) from estimated non-stationary gamma-distribution (i.e., gamma-parameters depend on the predictor and thus, vary from day-to-day); 2) compute quantiles of gamma-distribution with stationary parameters for these probabilities of a non-stationary distribution; 3) plot these quantiles against quantiles of stationary gamma-distribution for theoretical probabilities: $(1:n)/(n+1)$.

parameters. Note that the values of model and observation are shifted due to the standardization, depending on the strength of the predictor.

Improvements by the VGLM gamma compared to the predictor can be seen in most examples ranging from Scandinavia to the Mediterranean and from the Atlantic coast to eastern Europe in both seasons. However, in some locations the quantiles modeled by the VGLM gamma compare well to station observations (at least in Malaga better than the predictor) up to a certain quantile (e.g., Sibiu: $\sim$12 mm d$^{-1}$ and Malaga: $\sim$42 mm d$^{-1}$ in DJF) while there is a wet bias for intensities of the higher quantiles. It has been verified that precipitation at these locations is gamma-distributed (not shown). To understand this model behavior we analyze the predictor-predictand relationship of both observations and VGLM in Fig. 7 for DJF and appendix B2 for JJA. Circles are the observed gridded against point scale precipitation intensities, showing the spread of point scale predictands for a given grid scale predictor. The lines represent the 10 %, 25 %, 50 % (median), 75 %, 90 % and 95 % quantiles of the VGLM gamma model as a function of the predictor. This function of course fits best in the range where most of the values used to estimate the relationship are. For instance, in Sibiu (Malaga) for higher predictor values (Sibiu: >15 mm d$^{-1}$, Malaga: >42 mm d$^{-1}$) the predictands are around or below the 25 % (50 %) quantile of the model, and thus, simulated systematically too high by the VGLM. In both cases the bulk of the distribution is well captured however. This problem is also visible at other stations, e.g., Dresden or Karasjok. In JJA it is even more pronounced (appendix B2), particularly in Dresden and Sibiu where the high predictor values are even below the modeled 10 % quantile. These examples indicate that the VGLM basically allows for three different generalized linear relationships between the predictor and the parameters of the gamma distribution: concave (i.e., Brocken DJF), straight (i.e., San Sebastian DJF) or convex (i.e., Malaga DJF). No changes from lower to higher quantiles between these three types are possible. In some locations this appears to be not flexible enough to capture the true relationship which can be non-linear. A more flexible relationship that allows for a changed model behavior for higher values could improve the results but comes along with the risk of overfitting. Additionally, in eastern Europe the station density included in E-OBS is low[3]. Hence, in the E-OBS-gridbox closest to Sibiu, there may be only very few (one or two) stations included, implying most likely a misrepresentation of gridbox-precipitation. This problem affects the calibration of the model where E-OBS is used as reference as well as simulations employing E-OBS or precipitation that is corrected to E-OBS as predictor. We do not show results of the cross-validation here as the described problems with the VGLM in some locations are already present when repredicting the calibration period where the skill should be higher than in a cross-validation setup where a period is predicted that is not part of the calibration period. This clearly highlights deficiencies in the model for these locations.

In both DJF (Fig. 7) and JJA (appendix B2) Sonnblick and Brocken show a concave function whereas the function in the other example-stations is generally convex. The rain gauges at Sonnblick and Brocken are on top of the respective mountain. Although their climate is quite different as Sonnblick is a high mountain in the Alps (altitude: 3106 m) whereas the Brocken is the highest mountain in the northern German low mountain range Harz (altitude: 1142 m) they have an exposed position, coming along with high variability, in common. These results show that the VGLM gamma is capable to model the scale relationship for such exposed places of high variability quite well.

---

[3]For station density of actual E-OBS versions refer to the ECA&D website: http://www.ecad.eu/dailydata/datadictionary.php

[Figure 6 about here.]

[Figure 7 about here.]

### 5.2.3 Evaluation of the Combination of Steps 1 & 2: Bias Correction & Downscaling vs. Station

In the combined model the VGLM gamma, calibrated against E-OBS, is applied to the predictor selected in section 5.2.1
(Fig. 4). Here we evaluate precipitation simulated by predictor&VGLM with station observations as reference, and compare
it to the uncorrected RCM-simulated precipitation and to the $QM_{grid}$-corrected precipitation. The cross-validated time series'
are evaluated. For the VGLM the evaluation-criteria were computed for 100 realisations and then averaged.

To evaluate the predictor&VGLM-combined model we apply the same criteria as for the first step (bias correction, sec-
tion 5.2.1) but with station observations (i.e., point scale) as reference. The CvM-scores (a, b) and the percentage of simulated
values that are higher than the 95th percentile of the observations on wet days (%sim > perc95$_{obs}$, c, d) for the selected best
model based on the CvM-criterion are shown in Fig. 8, and summarized by boxplots for the European subdomains in Fig. 9.
QQ-Plots for example stations are provided in Figs. 10 for DJF and in appendix B3 for JJA. Precipitation is improved in most
cases by (parts of) our method. The uncorrected RCM (Fig. 8, triangles) is only selected at 8 (7) stations in DJF (JJA). Yet,
even if the RCM is selected the other models do not necessarily perform much worse such as in Stornoway in DJF (Fig. 10)
or in Malaga in JJA (appendix B3). The predictor&VGLM model (plotted as squares) is selected by CvM 25 times (45 times)
in DJF (JJA). The more frequent selection of the VGLM in JJA compared to DJF is likely related to the dominant underlying
mechanism, i.e., in summer there are many small scale convective precipitation events whereas in winter precipitation is mainly
caused by large scale weather systems.

The CvM-values of the selected model (Fig. 8a, b) indicate that the cdf-bias is again generally lower in JJA than in DJF,
and for DJF lowest in the Mediterranean region. In eastern Europe and Scandinavia in DJF the VGLM is only rarely selected
– in these regions the $QM_{grid}$-corrected time series which is on grid scale is mostly selected although the reference-cdf is on
point scale (Fig. 8a). This might be due to problems with the VGLM gamma as explained in section 5.2.2. The rather large
cdf-bias in ME, SC and EA in DJF could hence be related to the remaining scale gap as the $QM_{grid}$-corrected time series is
not expected to correctly represent the point scale. The QQ-Plot of Sibiu in DJF (Fig. 10) illustrates this problem. The higher
$QM_{grid}$-corrected quantiles are as expected too low and the VGLM fails at this station in DJF (see also section 5.2.2). Finding
an adequate stochastic model to bridge the scale gap might improve the representation of precipitation in such cases. Also in
JJA there are examples where the VGLM has not been selected but a suitable VGLM would likely further improve the results
(appendix B3, San Sebastian, Dresden & Karasjok). For, e.g., Brocken JJA and Sion DJF an improved VGLM may likely even
improve the result although the VGLM has been selected. However, finding the optimal model for all 86 stations is beyond the
scope of our study. The boxplots confirm again the good performance for DJF in the Mediterranean region (MD & IP), and also
in AL (Fig. 9a). The CvM-score and thus, the cdf-bias is again very low in JJA, indicating good performance of our method
with no pronounced difference between the European subregions (Fig. 9b). Yet, the sensitivity of the CvM-score is illustrated
by Stornoway in JJA (appendix. B3) as this example still yields suitable results despite the relatively high CvM-score.

The occurrence of heavy precipitation in the CvM-selected model is slightly overestimated in most subregions in DJF (Figs. 8c & 9c), though quite well represented in IP & FR (Fig. 9c). In JJA heavy precipitation occurence is quite well estimated (Figs. 8d & 9d). The median of most subregions is very close to 5 % (the "perfect" model would have exactly 5 %). However, some stations, particularly in EA, underestimate the occurrence of heavy precipitation. These are in most cases stations where the VGLM has not been selected, likely indicating problems with the VGLM and the remaining scale gap (see paragraph before & section 5.2.2).

Ideal performance of our combined model is illustrated in the example QQ-Plot of Malaga in DJF (Fig. 10), i.e., $QM_{grid}$ corrects the RCM-simulated precipitation on the same scale and the VGLM bridges the remaining scale gap, resulting in a good match of the observations. Sonnblick in DJF (Fig. 10) and JJA (appendix B3) and Brocken in DJF (Fig. 10) are also well performing examples. The QQ-Plot of San Sebastian in DJF (Fig. 10) shows the benefit of selecting the predictor by CvM as in this case the RCM is used as predictor for the VGLM. Here using the $QM_{grid}$-corrected time series may result in too high extremes. Sion in JJA (appendix B3) is another good example for the benefit of model selection where the RCM has been selected as predictor. Here the high VGLM-simulated quantiles are already overestimated in this setting and would likely be even higher should the $QM_{grid}$-corrected predictor be employed.

[Figure 8 about here.]

[Figure 9 about here.]

[Figure 10 about here.]

## 5.3 Intercomparison of all Models

In this section an intercomparison of all models (not only the selected best model from section 5.2) for all subregions is presented and compared to the classical application of $QM_{point}$. Figure 11 shows boxplots for the CvM-score. Generally the cdf-bias is lower in JJA than in DJF for all models, already for the uncorrected RCM (apart from BI). In the Mediterranean region (MD & IP) there is a very low cdf-bias in all models, indicating general good performance. The QM improves the cdf-bias in many regions with $QM_{grid}$ and $QM_{point}$ being similar in many cases. The effect of the VGLM depends on region and season. The representation of precipitation is generally improved by the VGLM in BI, IP, AL and MD in both seasons. However, in FR, ME, SC and EA in DJF the VGLM introduces biases. The bias increases from west to east (FR → ME → EA) with largest spatial variability in EA, extending to high CvM-values. For continental winter climate the used VGLM gamma model appears thus to be not the ideal model which suggests that in these regions it may be better to only correct the bias. This raises the question why the results become worse when statistical post-processing is applied. One potential reason for these problems with the VGLM in some regions is that the VGLM gamma is not flexible enough to capture the true predictor-predictand relationship if this relationship is non-linear as discussed in sections 5.2.2 & 5.2.3. The final downscaled marginal distribution may thus be wrong even though it was properly adjusted by the bias correction step. As the predictor-predictand relationship is always estimated such that it follows well the bulk of the distribution this problem occurs for predictand values at the very low

ends of the VGLM conditional distribution. Furthermore, particularly in EA and FR, E-OBS may be an inappropriate reference for calibration in both $QM_{grid}$ and VGLM due to low station density. Yet, in SC stations in E-OBS are relatively dense and thus, the bias introduced by the VGLM is in that case not attributable to E-OBS quality. In DJF SC has the highest RCM-bias among all subregions. This suggests a detailed evaluation of this high bias which is beyond the scope of our study however.

5                                                        [Figure 11 about here.]

To infer the performance of all studied models in estimating the occurrence of heavy precipitation boxplots for the percentage of simulated values that are higher than the 95th percentile of the observations on wet days (%sim > perc95$_{obs}$) for all models are provided in Fig. 12. Particularly in JJA the $QM_{grid}$ improves the occurrence of heavy precipitation but remains slightly too dry which is expected due to the remaining scale gap. The estimated occurrence of heavy precipitation is improved by the
VGLM in many cases, although generally slightly overestimated. The results of the VGLM and $QM_{point}$ are generally similar with the $QM_{point}$ being often slightly closer to the 5%-line and the VGLM slightly too wet. In AL the VGLM considerably improves the cdf-bias (Fig. 11f) and the occurrence of heavy precipitation (Fig. 12f) in both DJF and JJA compared to the uncorrected and $QM_{grid}$-corrected RCM. In SC in DJF one should be careful as although the occurrence of heavy precipitation is considerably improved by the VGLM (Fig. 12e) it introduces biases when the whole cdf is evaluated (Fig. 11e) and is thus
not recommended. Concerning heavy precipitation occurrence our model shows a similar behavior for all subregions in JJA and for IP also in DJF (Fig. 12). The $QM_{grid}$ bias correction improves the representation but remains too dry. The dry bias is then eliminated by the VGLM though to slightly too many "extremes". This model behavior as exhibited in JJA is exactly what would be expected due to the scale gap between gridded and point scale. Due to more small scale convective extremes this scale gap has a larger impact in summer whereas in winter most extremes are caused by large scale weather systems that are
generally better represented by the gridbox scale, also in coarser resolutions. While the cdf-bias and the occurrence of heavy precipitation reveal how well properties of the precipitation distribution are represented they do not allow to draw conclusions about the predictive power of the model.

[Figure 12 about here.]

To infer whether our model has predictive power we cannot assess temporal correspondence compared to observations as
in Wong et al. (2014) and Eden et al. (2014) because we use an RCM that is not nudged and even though driven with perfect boundary conditions (reanalysis) this is not a clean pairwise setup. Instead, we evaluate spatial autocorrelation which is the correlation of a variable with itself in geographical space. This allows to evaluate whether the model correctly reproduces daily spatial autocorrelations and thus, the spatial extent of precipitation patterns including its variability in time compared to observed precipitation. In Fig. 13 correlograms of the cross-validated time series' of all models (RCM, $QM_{grid}$, $QM_{point}$,
100 VGLM-realisations) and station observations as reference are provided. The spatial autocorrelation of QM-bias corrected precipitation decays very similar to uncorrected RCM-precipitation and shows thus only little improvement of spatial autocorrelation compared to point scale observations. Differences between $QM_{grid}$ and $QM_{point}$ are negligible. This confirms that the QM-approach is not capable to model small scale variability, and a stochastic model is thus needed to bridge the scale

gap. The spatial autocorrelation of VGLM-downscaled precipitation decays more similar to the station observations than the QM-corrected or uncorrected RCM, particularly in JJA. The spatial dependence is thus improved by the stochastic downscaling step. The long decorrelation length in DJF is underestimated by our stochastic, single-site model, which indicates a slightly too strong noise component. A spatial model considering more than one station or including more physical based predictors (i.e.,

sea level pressure) might improve the predictive power of our model in DJF.

[Figure 13 about here.]

## 6   Conclusions

We introduced the concept of a combined statistical bias correction and stochastic downscaling method for precipitation. We thereby extend the stochastic Model Output Statistics (MOS)-approach developed by Wong et al. (2014) beyond nudged

simulations to free running GCM/RCM-simulations. We applied our method to precipitation simulated by the RCM KNMI-RACMO2 driven with ERA-Interim boundary conditions within the EURO-CORDEX framework. As the RCM is driven with reanalysis we only correct RCM-biases. Our method corrects the "drizzle effect" (i.e., too many wet days), too low precipitation values in the rain shadows caused by not enough windward air masses crossing the mountain range ("location bias", Maraun and Widmann, 2015), and precipitation intensity. To correct the "drizzle effect" we increased the wet day threshold such

that the number of wet days (closely) matches the gridded observations with a threshold of 0.1 mm d$^{-1}$ (Maraun, 2016). To overcome the "location bias" we selected the RCM-gridbox that best represents the climate in the respective gridbox of the gridded observations (Maraun and Widmann, 2015). Note that when transferring the approach to free-running simulations this grid box selection step has to be calibrated with a reanalysis-driven simulation of the RCM to ensure temporal correspondance. Consequently, only the location bias caused by the RCM is corrected. How a potential location bias of the driving GCM may

affect the results should be analyzed in future work. Precipitation intensities were corrected by a parametric quantile mapping (QM) approach between RCM and gridded observations on the same spatial scale. As precipitation is highly variable in space and time not all variability can be explained by the gridbox scale (Maraun, 2013a). To bridge the gap between gridbox and point scale we applied a stochastic regression-based model. For evaluation we adopted the experimental framework of VALUE (Maraun et al., 2015). In this context, we applied our method to 86 example rain gauges across Europe representing different

climates, and carried out a five-fold cross-validation for the time period 1979-2008. Both steps of the combined method were evaluated individually and combined. A comparison to classical QM between RCM and point scale is also provided.

The proposed parametric model structure appears not to be the optimal choice for all considered stations. Yet given that the aim of our study is a proof of concept, the idenfification of an optimal model for all individual cases would be beyond the scope of this work. Nevertheless, where our implementation is not adequate we provide suggestions for improvements within

the presented framework. Our specific implementation for the QM-bias correction (first step) of wet day intensities employs the mixture distribution of a gamma distribution for the precipitation mass and a generalized Pareto (GP) distribution for the extreme tail (Frigessi et al., 2002; Vrac and Naveau, 2007). The stochastic regression-based model for downscaling (second step) was calibrated between observations on gridded and point scale, and then transferred to bias corrected RCM-simulated

precipitation. This corresponds to a perfect prog (PP)-approach. The regression model consists of a logistic regression to model wet day probabilities and a vector generalized linear model (VGLM) predicting the parameters of a gamma probability distribution for precipitation intensities. The QM-corrected time series (first step) was used as predictor for downscaling (second step) if it improves the representation of precipitation compared to the uncorrected RCM. Thus, we selected the predictor based on the lower cdf-bias by applying the Cramér-von Mises (CvM)-criterion with the gridded E-OBS dataset as reference.

Precipitation was in most cases improved by (parts of) our combined method across different European climates, to what extent depends on region and season though. The method generally performs better in JJA than in DJF and in DJF best in the Mediterranean region with a mild winter climate and worst for continental winter climate in mid & eastern Europe or Scandinavia. Seasonal and regional differences depending on the underlying mechanism have already been reported for resolution dependence of extreme precipitation in GCMs (Volosciuk et al., 2015) and RCMs (Prein et al., 2013; Meredith et al., 2015). Hence, for a good representation of precipitation extremes the complexity of the model can be chosen at each step of the modeling cascade based on the underlying mechanism in order to use computational resources efficiently.

Although our bias correction (first step) improved simulated precipitation for many locations in both seasons wet biases may remain even after bias correction, particularly for continental winter. In agreement with our results large improvements by bias correction over the Alps, Spain and France have been reported by Dosio and Paruolo (2011). Yet, in contrast to our results these authors also obtain good results for middle and eastern Europe where we find persisting biases even after bias correction. In the cases where the quantile mapping approach does not improve RCM simulated precipitation another transfer function might be more suitable. Choosing between different parametric transfer functions as proposed by Piani et al. (2010) could improve the results. By employing a quantile mapping approach we presumed both a stationary statistical relationship and stationary cdfs that also apply in a changed future climate. However, in a climate change context RCM-simulated trends in the cdf are modified by applying such statistical post-processing. For cases where the GCM/RCM simulates plausible climate change trends the CDF-t concept suggested by Michelangeli et al. (2009) and Vrac et al. (2012) might be an appropriate framework. In their concept the correction function explicitly accounts for future trends in the RCM-simulated distribution. Thereby simulated trends in all moments are approximately preserved after bias correction. For instance, regions where an increase in extreme precipitation accompanied by a decrease in mean precipitation is projected (e.g., in Central European summer, Christensen and Christensen, 2003; Maraun, 2013b) these trends might be better represented by employing a CDF-t method. Yet, in this study we have not employed this variant as in our setting the validation period is too short to achieve an appropriate fit of the future mixture distribution. Quantifying the differences between the quantile mapping approach we employed here and a CDF-t approach is left for future work when our combined method will be applied to climate change scenarios.

The stochastic downscaling (second step) improves the estimated occurrence of heavy precipitation in many regions but introduces biases in continental winter climate. Furthermore, spatial autocorrelation in JJA is improved by the VGLM showing the importance of randomization in the framework of downscaling as already pointed out by, e.g., von Storch (1999) and Maraun (2013a). Moreover, when downscaling climate change scenarios the randomization component of the VGLM that adds small scale unexplained variability does not modify trends in contrast to purely deterministic methods, e.g., QM (Maraun, 2013a). Yet, the deterministic part of the VGLM that corrects systematic local effects (e.g., lee/windward side of a mountain)

alters the pdf, and may thus also change trends. The stochastic downscaling-step is more important in JJA than in DJF for both estimation of heavy precipitation occurence and spatial autocorrelation. This can be attributed to the different underlying main mechanism for heavy precipitation. In summer heavy precipitation is often caused by small scale convective events whereas in winter large scale weather systems dominate. Hence, there is less small scale variability unexplained by the gridbox in

DJF. In DJF spatial autocorrelation is slightly underestimated by the VGLM which is likely related to the long decorrelation length of precipitation in winter that is not correctly represented in our single-site model, indicating a slightly too strong noise component. An extension of our method to a multi-site model and/or including more physical based predictors (i.e., sea level pressure) would likely improve this feature and can be subject of future work. A possible extension to multi-variate or full fields might be based on copulas (e.g., Ferraris et al., 2003; Schoelzel and Friederichs, 2008; Bárdossy and Pegram, 2009) or random

cascade models (Thober et al., 2014). A good representation of the mild climate on the British Isles is consistent with Wong et al. (2014) and Eden et al. (2014). In France, Mid-Europe, eastern Europe and Scandinavia in DJF the VGLM introduces biases, raising the question why the results become worse when statistical post-processing is applied. Particularly in France and eastern Europe the E-OBS gridded observational dataset may be an unreliable reference for model calibration for both the QM and the VGLM due to low station density. The "true" resolution of E-OBS in these regions might be coarser than the resolution

it is gridded to. This highlights that the applicability of our method is limited to regions where high quality gridded datasets are available. Yet, a detailed evaluation of the sensitivity of our method to station density in the gridded dataset is beyond the scope of this study. The bias introduced by the VGLM generally increases from west to east, and thus, from maritime to continental winter climate. However, in Scandinavia the VGLM also introduces biases even though station density is high. This indicates that although the quality of the E-OBS data may contribute to these problems it can not be identified as the main source of

error. It is rather one potential reason among others. For instance, in some cases the generalized linear relationship between the predictor and the parameters of the gamma distribution appears to be not flexible enough to capture the true predictor-predictand relationship which can be non-linear, particularly in but not restricted to continental winter climate. In these regions there may be a more adequate parametric relationship than our specific implementation. Problems with the current implementation may be related to, e.g., the linear structure of the model or the choice of the link function. For instance, another distribution in the

VGLM (e.g., mixture model), splines as applied in Maraun et al. (2011b) or a vector generalized additive model (VGAM, Yee and Wild, 1996) are potential approaches. Yet, employing a more complex model also comes along with the risk of overfitting. Finding the optimal model for each of the analyzed stations is beyond the scope of this study however.

The varying performance of our specific implementation clearly shows that bias correction and downscaling methods should be reevaluated when transferring it to locations with different climatic conditions. In some regions a specific implementation

different from the one we used is required. We recommend our model in summer for all studied regions. Yet, in winter it should only be used for the British Isles, the Alps, the Mediterranean region and the Iberian peninsula but not for continental winter climates (Scandinavia, Mid-Europe and eastern Europe) and France. While the stochastic downscaling step (VGLM) is very important to represent spatial autocorrelation in summer it is less important in winter where the applicatin of solely the bias correction step might be sufficient. The concept can generally be extended to a wide range of method combinations.

Transferring this concept to other climate variables should in principle be possible. Our specific implementation should be

applicable to any gamma-distributed variable. However, our approach has so far only been evaluated for precipitation. Thus, users need to evaluate the model for the particular variable at the chosen location when transferring it.

We developed our model in present day climate. In a climate change context the model does not explicitly modify climate trends on a physical basis. Our model is thus only applicable where changes are correctly simulated by the GCM/RCM. For instance, changes in the dynamics of local extreme convective events in summer that need even higher resolution up to convection-permitting simulations (e.g., Kendon et al., 2014; Chan et al., 2014; Meredith et al., 2015) will also not be represented after statistical post-processing is applied. Bias correction and (dynamical and statistical) downscaling of precipitation is only applicable if the large scale patterns and changes therein are simulated reasonably by the driving GCM (Eden et al., 2012; Hall, 2014). Therefore, when transferring our method to a GCM or GCM-driven RCM the relevant processes for precipitation in the studied region need to be correctly simulated. For instance, biases in simulated precipitation related to biases in the storm track (Chang et al., 2012), El Niño-Southern Oscillation (ENSO; Zhang and Sun, 2014), the monsoon (Hasson et al., 2013) or persistent weather regimes (Petoukhov et al., 2013; Palmer, 2013) cannot be statistically corrected in a physical sensible way.

The general concept of combining two methods and thereby separating bias correction (MOS) and downscaling (PP) into two steps is a powerful approach as it benefits from the respective methodological advantages. Additionally, the strength of this two-step method is that the best combination of methods can be selected. This implies that the concept can be extended to a wide range of method combinations.

## Appendix A: Technical Details for Bias Correction Implementation and Model Selection

### A1 Technical Details for Model Implementation

A non-zero wet day threshold assigns zero probability density to all intensities between zero and the threshold, resulting in a misfit of the gamma distribution (Wong et al., 2014). To avoid this we shift precipitation on all wet days by subtracting the threshold for calibration. The estimated distribution is subsequently shifted back by the threshold.

Numeric instabilities in the estimation of the mixture cdf may in rare cases result in a discontinuous cdf (Fig. 14a). In these cases we interpolate linearly between the continuous probabilities surrounding the discontinuity. The example cdf in Fig. 14a illustrates that this procedure is a reasonable estimation for these quantiles. If the cdf does not "jump back" as in Fig. 14a but continues as illustrated in Fig. 14b the model has to be sorted out as there is no straightforward possibility to handle this artifact caused by numerical instability. Yet, the latter case only occurs extremely seldom.

[Figure 14 about here.]

### A2 Technical Details for Model Selection

The AIC performs best for the part of the distribution where most of the values are. Hence, a good fit for the bulk of the distribution might include large biases in the extremes and still have the lowest AIC (example: Fig. 14c). To avoid such a model choice with unreasonable high extremes we introduce a criterion based on the extremes to sort out mixture model

fits yielding too high extremes before AIC-model selection is applied. This criterion is based on a comparison between the 100 season return level estimated by the mixture model (RL100S$_{mixture}$) and the 95 % confidence interval of the RL100S$_{GP}$ estimated by the GP distribution only. The RL100S$_{GP}$ and the corresponding 95 % confidence interval are estimated according to Coles (2001). This criterion is applied differently for F$_{obs}$ and F$_{RCM}$ considering the respective relevant quantity for the correction function. For F$_{obs}$ this criterion is based on the return level itself whereas for F$_{RCM}$ the probability for the return level is considered. In particular, for F$_{obs}$ the RL100S$_{mixture}$ must not exceed the 95 % confidence interval of the RL100S$_{GP}$. For F$_{RCM}$ the mixture model-probability ($p_{mixture}$) for the RL100S$_{GP}$ must not exceed $p_{GP}$ for the 95 % confidence interval of RL100S$_{GP}$. Furthermore, $p_{mixture}$ for the 95 % confidence interval of the RL100S$_{GP}$ must not be very close to 1 (i.e., >1–1e-15) as a reasonable extrapolation to potentially higher values under climate change would not be possible in that case.

## Appendix B:  Additional Results

### B1    Step 1: Bias Correction

[Figure 15 about here.]

### B2    Step 2: Downscaling

[Figure 16 about here.]

[Figure 17 about here.]

### B3    Combination of Steps 1 & 2: Bias Correction & Downscaling

[Figure 18 about here.]

*Author contributions.*  DM had the initial idea for this combined method. CV implemented the method and performed the evaluation with help from MV and DM. All authors discussed details of the implementation and the results. CV prepared the manuscript with contributions from all co-authors.

*Competing interests.*
The authors declare that they have no conflict of interest.

*Acknowledgements.*  We thank the KNMI for producing and making available their model output. We acknowledge the E-OBS dataset from the EU-FP6 project ENSEMBLES (http://ensembles-eu.metoffice.com) and the data providers in the ECA&D project (http://eca.knmi.nl). We thank S. Kotlarksi, S. Hagemann and one anonymous reviewer for comments on the manuscript. The analysis was carried out with R, using the packages evir, ncdf, MASS, stats, stats4, fields, aspace, ncf and rworldmap. This study was funded by the EUREX project of the Helmholtz Association (HRJRG-308) and the PLEIADES project of the Volkswagen Foundation (Grants 85423 and 85425). C. Volosciuk has received a Short-Term Scientific Mission Grant from the EU COST Action ES1102 VALUE.

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

## List of Figures

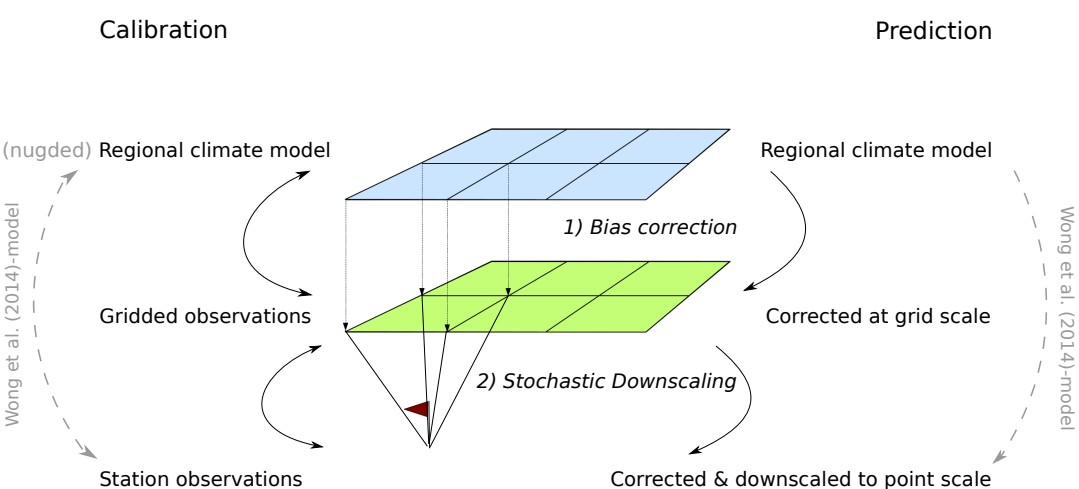

**Figure 1.** Schematic of (black) our combined statistical bias correction and stochastic downscaling model, and (grey) the Wong et al. (2014)-model.

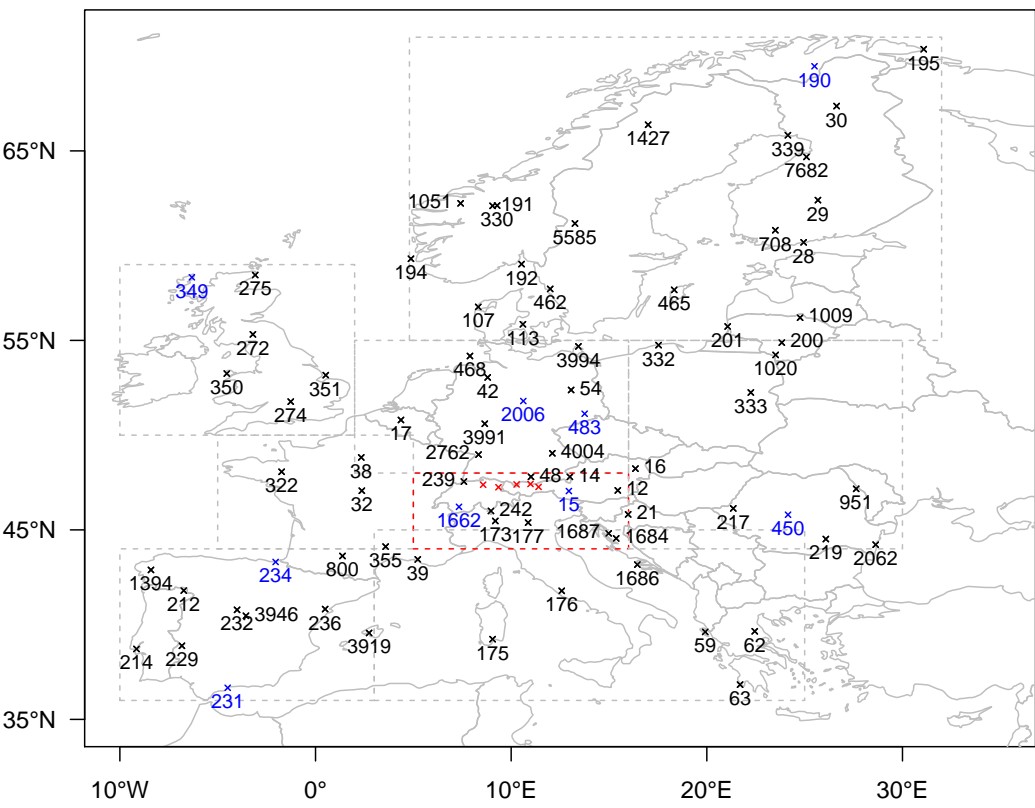

**Figure 2.** Location and IDs of used rain gauges from ECA&D. IDs of red marked stations from left to right: 244, 243, 4002, 58, 13. Stations for detailed analysis are marked blue. Dashed lines represent European subdomains for analysis as defined by PRUDENCE-project (Christensen and Christensen, 2007): the British Isles (BI), the Iberian Peninsula (IP), France (FR), Mid-Europe (ME), Scandinavia (SC), the Alps (AL, dashed red line), the Mediterranean (MD) and eastern Europe (EA)

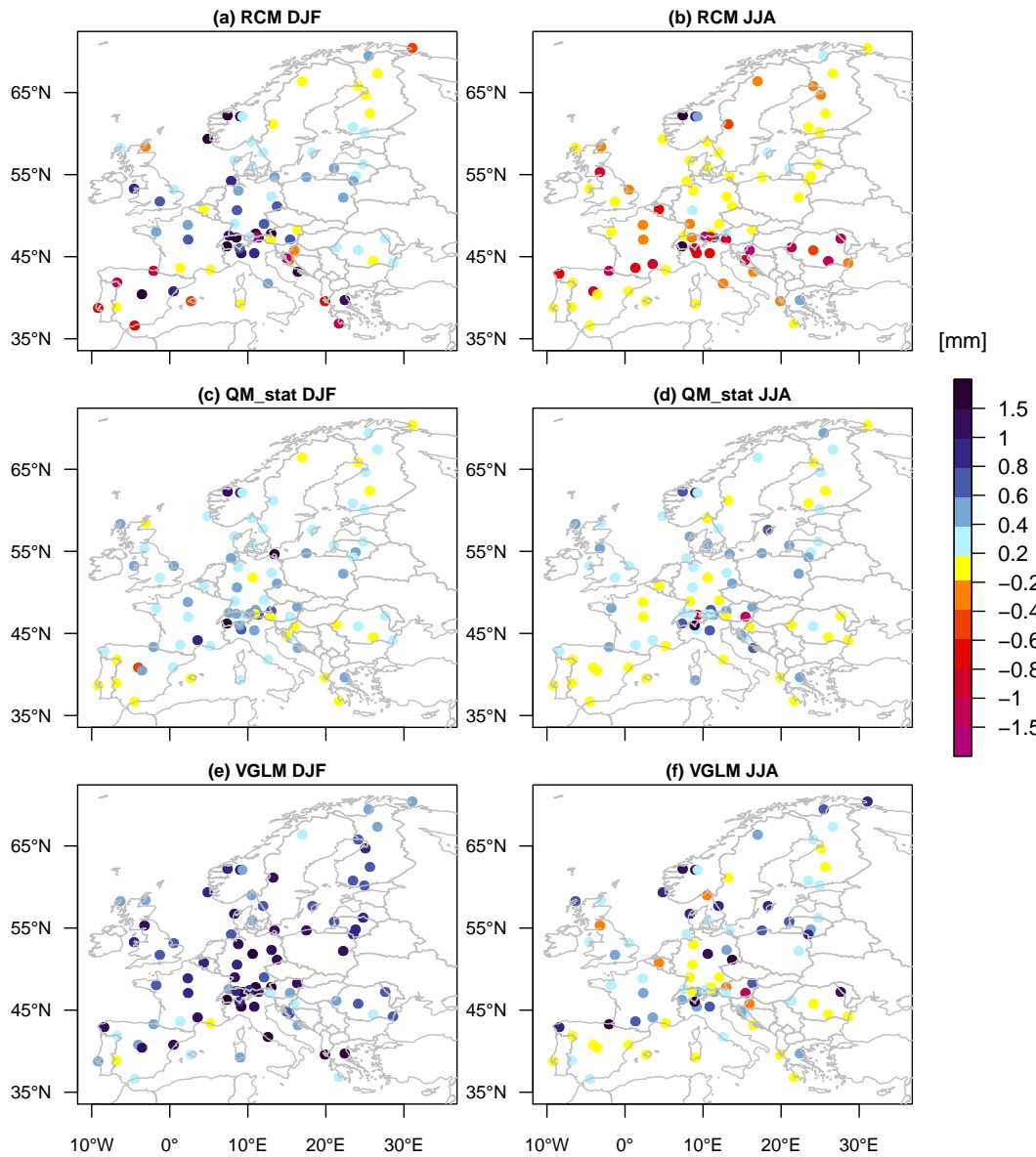

**Figure 3.** Mean bias. (a, b) Uncorrected RCM, (c, d) $QM_{point}$-corrected RCM to the point scale, (e, f) selected predictor (RCM or $QM_{grid}$ corrected RCM)&VGLM. Reference is station data.

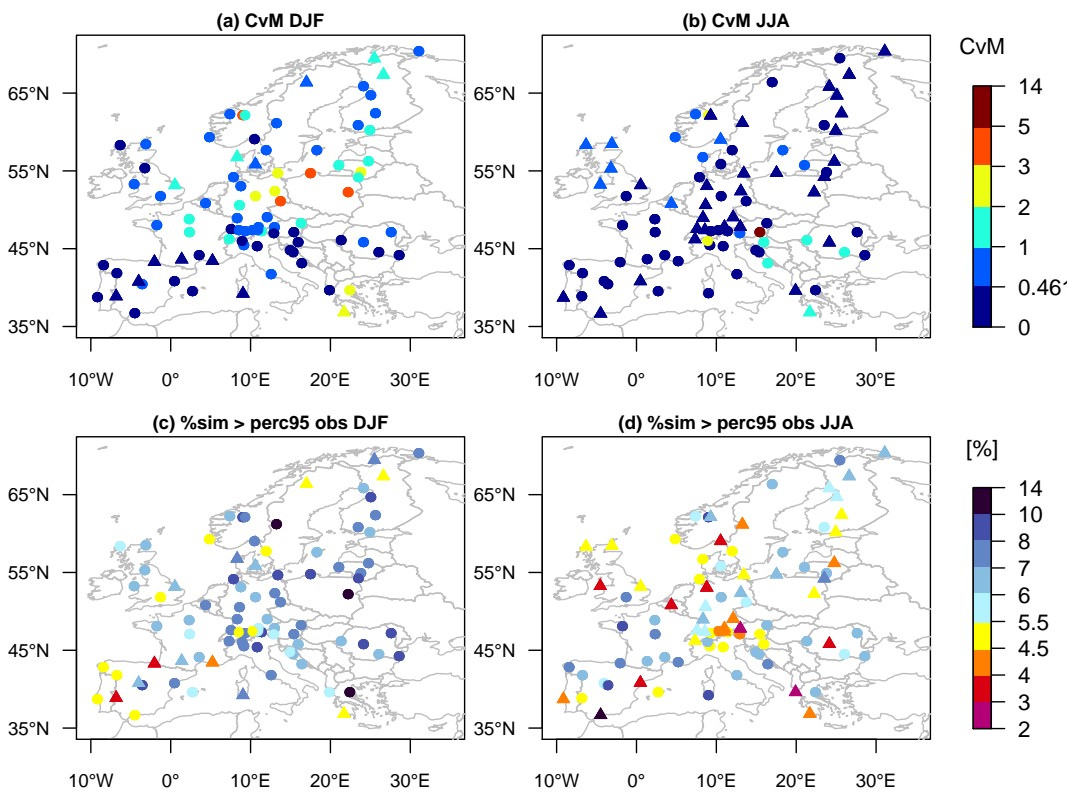

**Figure 4.** Step 1: Bias correction to grid scale. (a, b) CvM-score for selected cross-validated predictor against E-OBS. Threshold for values under which the model-CDF is not statistically significantly different at the 95%-level from reference-CDF: 0.461. (c, d) Percentage of wet days in CvM-selected cross-validated predictor exceeding the 95th percentile of wet days in E-OBS (%sim > perc95$_{obs}$). Selected model: circles: QM$_{grid}$-corrected RCM, triangles: uncorrected RCM.

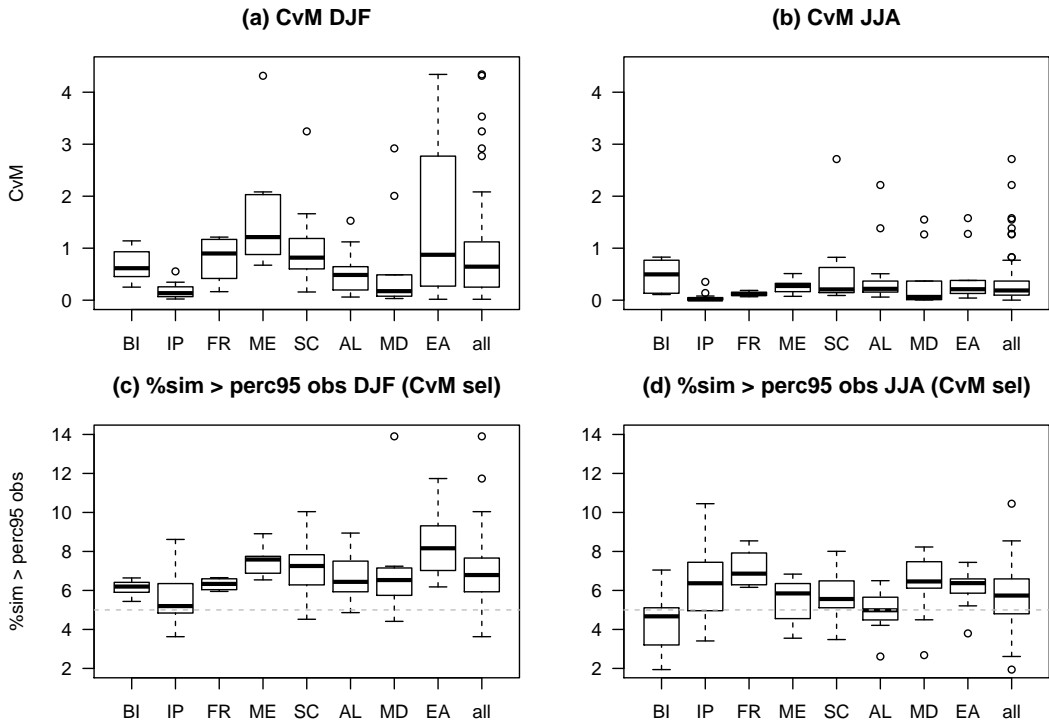

**Figure 5.** Step 1: Bias correction to grid scale. Boxplots of (a, b) CvM-score and (c, d) percentage of simulated wet days exceeding the observed 95th percentile (%sim > perc95$_{obs}$) for CvM-selected cross-validated predictor in European subdomains: British Isles (BI), Iberian Peninsula (IP), France (FR), Mid-Europe (ME), Scandinavia (SC), Alps (AL), Mediterranean (MD) and eastern Europe (EA). Outlier out of range in (b) AL & all: 12.95.

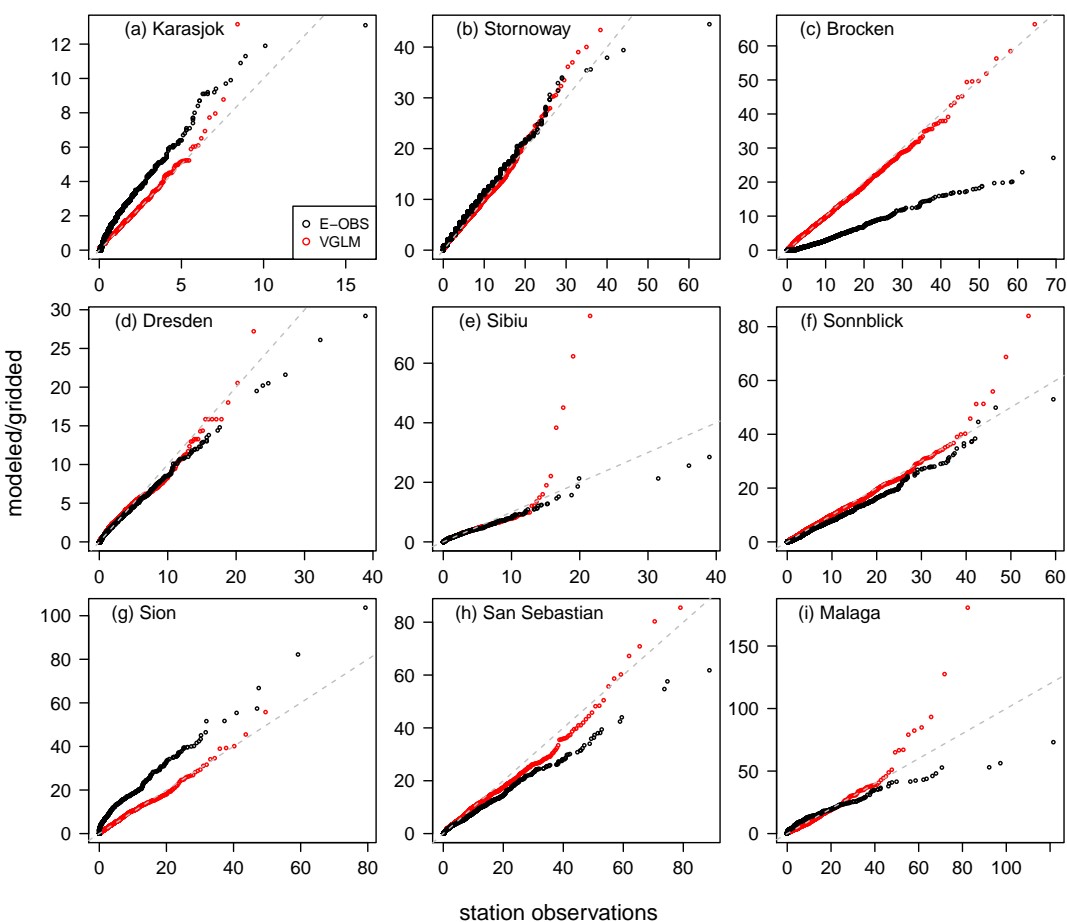

**Figure 6.** Step 2: Downscaling. QQ-Plots for example stations in DJF. VGLM gamma standardized to stationary gamma distribution fitted to observed wet day intensities between gridded and point scale precipitation observations (mm d$^{-1}$). (a) Karasjok, (b) Stornoway, (c) Brocken, (d) Dresden, (e) Sibiu, (f) Sonnblick, (g) Sion, (h) San Sebastian and (i) Malaga.

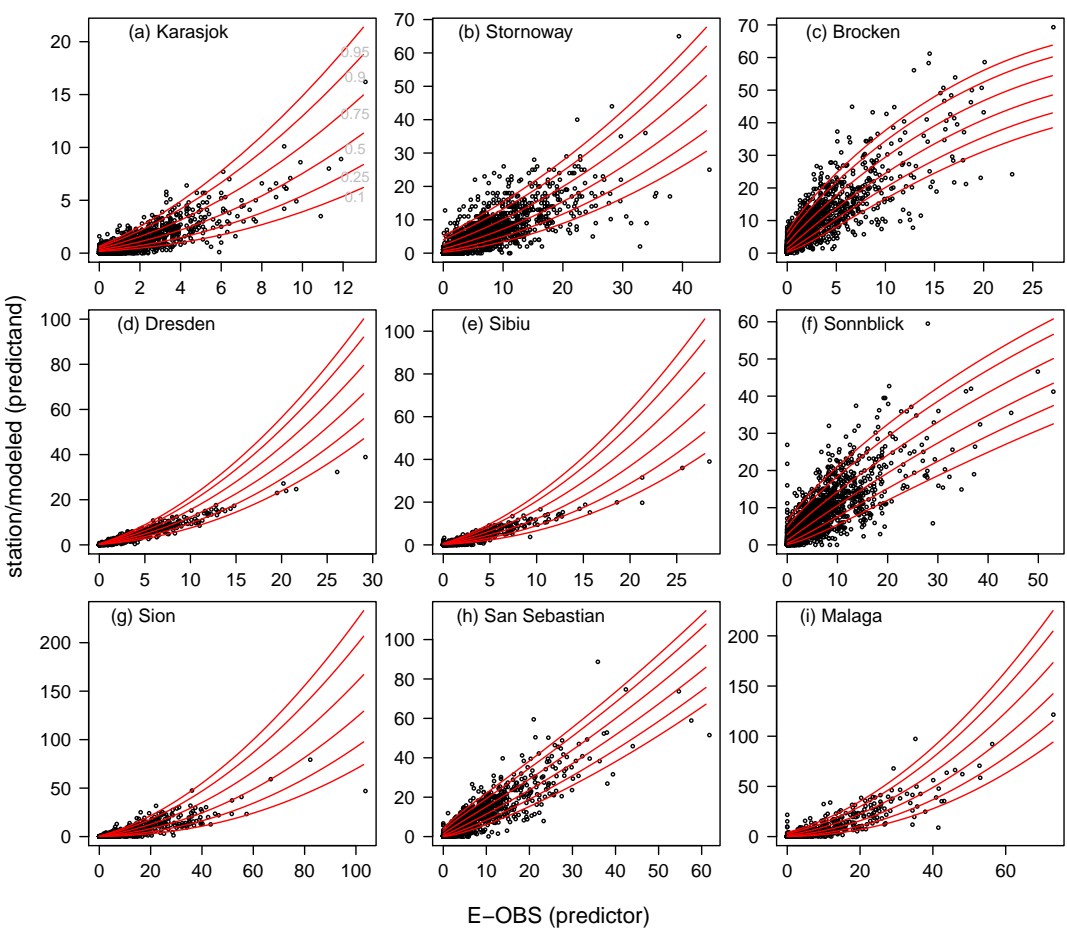

**Figure 7.** Step 2: Downscaling. Estimated relation between gridded and point scale precipitation observations for example stations in DJF. VGLM gamma where both parameters depend on the predictor fitted to observed wet day intensities. The predictor is E-OBS. Circles: observed precipitation intensities (mm d$^{-1}$), lines: 0.1, 0.25, 0.5, 0.75, 0.9 and 0.95 modeled quantile (mm d$^{-1}$). (a) Karasjok, (b) Stornoway, (c) Brocken, (d) Dresden, (e) Sibiu, (f) Sonnblick, (g) Sion, (h) San Sebastian and (i) Malaga.

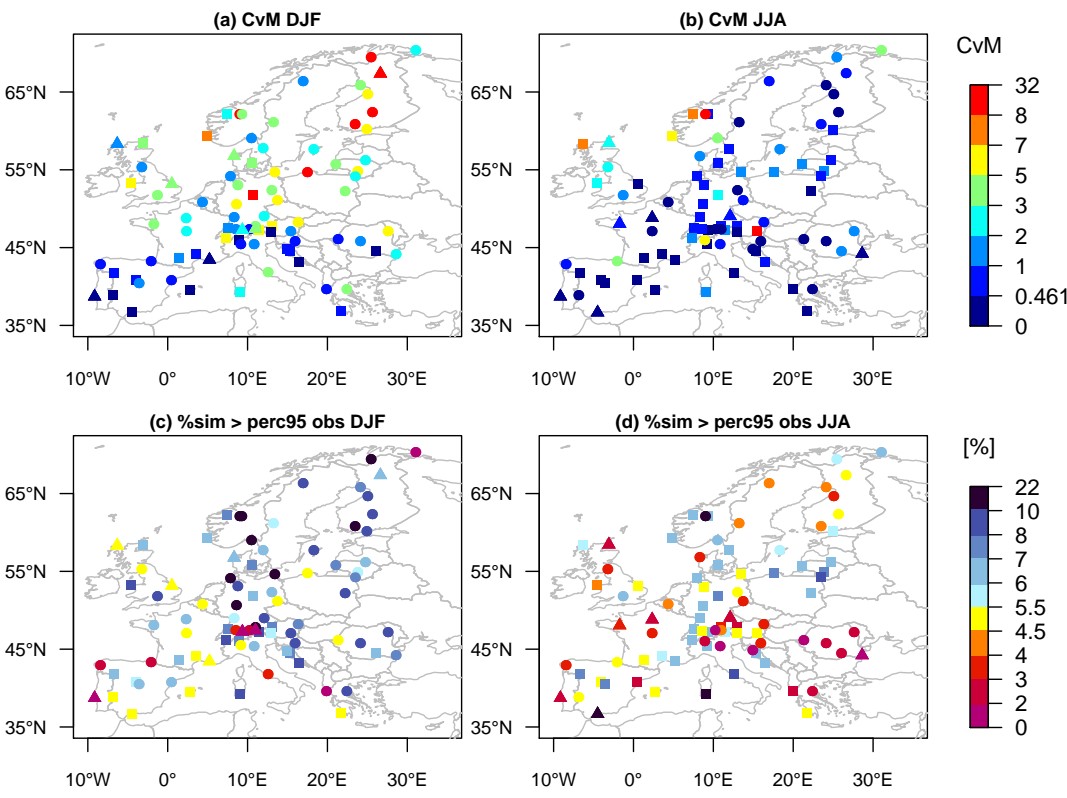

**Figure 8.** Step 1 & 2: Combined model. (a, b) CvM-values for selected cross-validated model. Threshold for values under which the model-CDF is not statistically significantly different at the 95%-level from reference-CDF: 0.461. (c, d) Percentage of cross-validated model values exceeding the 95th percentile of station observations (%sim > perc95$_{obs}$) for cross-validated CvM-selected model. For the VGLM the criteria were computed for 100 realisations and then averaged. Selected model: squares: predictor&VGLM, circles: QM$_{grid}$-corrected RCM, triangles: uncorrected RCM. Note the different color scales than in Fig. 4.

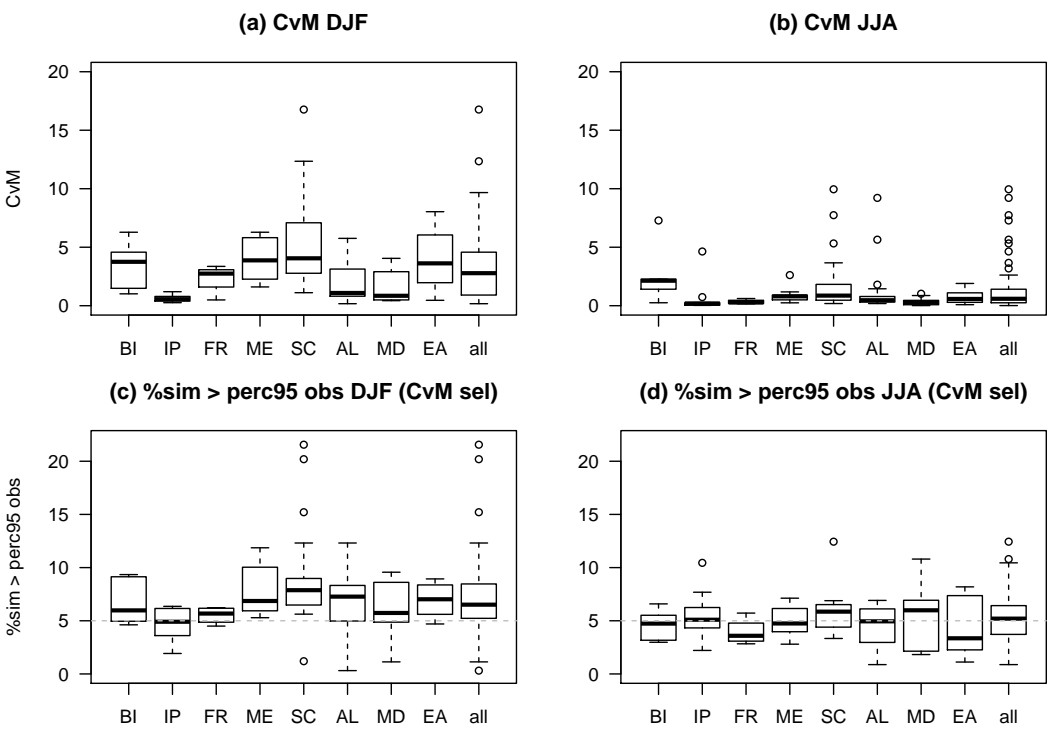

**Figure 9.** Step 1 & 2: Combined model. Boxplots of (a, b) CvM-score and (c, d) percentage of simulated wet days exceeding the observed 95th percentile (%sim > perc95$_{obs}$) for cross-validated CvM-selected model. Regions: British Isles (BI), Iberian Peninsula (IP), France (FR), Mid-Europe (ME), Scandinavia (SC), Alps (AL), Mediterranean (MD) and eastern Europe (EA). Note the different scales of the y-axes than in Fig. 5. Outliers out of range in (a) ME & all: 22.19; SC & all: 27.12 & 31.35.

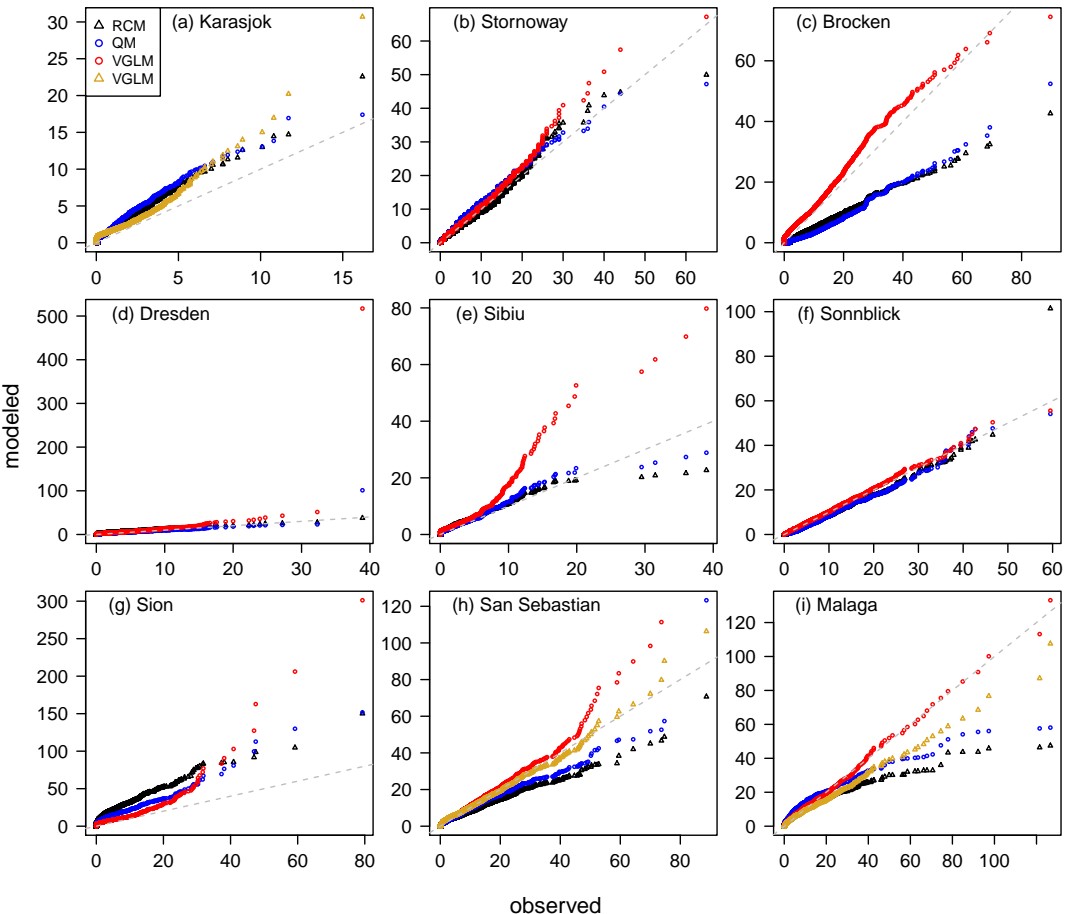

**Figure 10.** QQ-Plots for example stations of different models (cross-validated) against station observations for DJF (mm d$^{-1}$). (a) Karasjok, (b) Stornoway, (c) Brocken, (d) Dresden, (e) Sibiu, (f) Sonnblick, (g) Sion, (h) San Sebastian and (i) Malaga. For the VGLM the quantiles (i.e., sorted time series) of 100 realisations are averaged. Predictor for VGLM as selected by CvM-criterion: (red circles) QM$_{grid}$-bias corrected RCM, (brown triangles) uncorrected RCM; For examples to illustrate model performance and predictor selection (San Sebastian and Malaga) the VGLM is plotted for both predictors. Selected predictor: San Sebastian: RCM, Malaga: QM$_{grid}$.

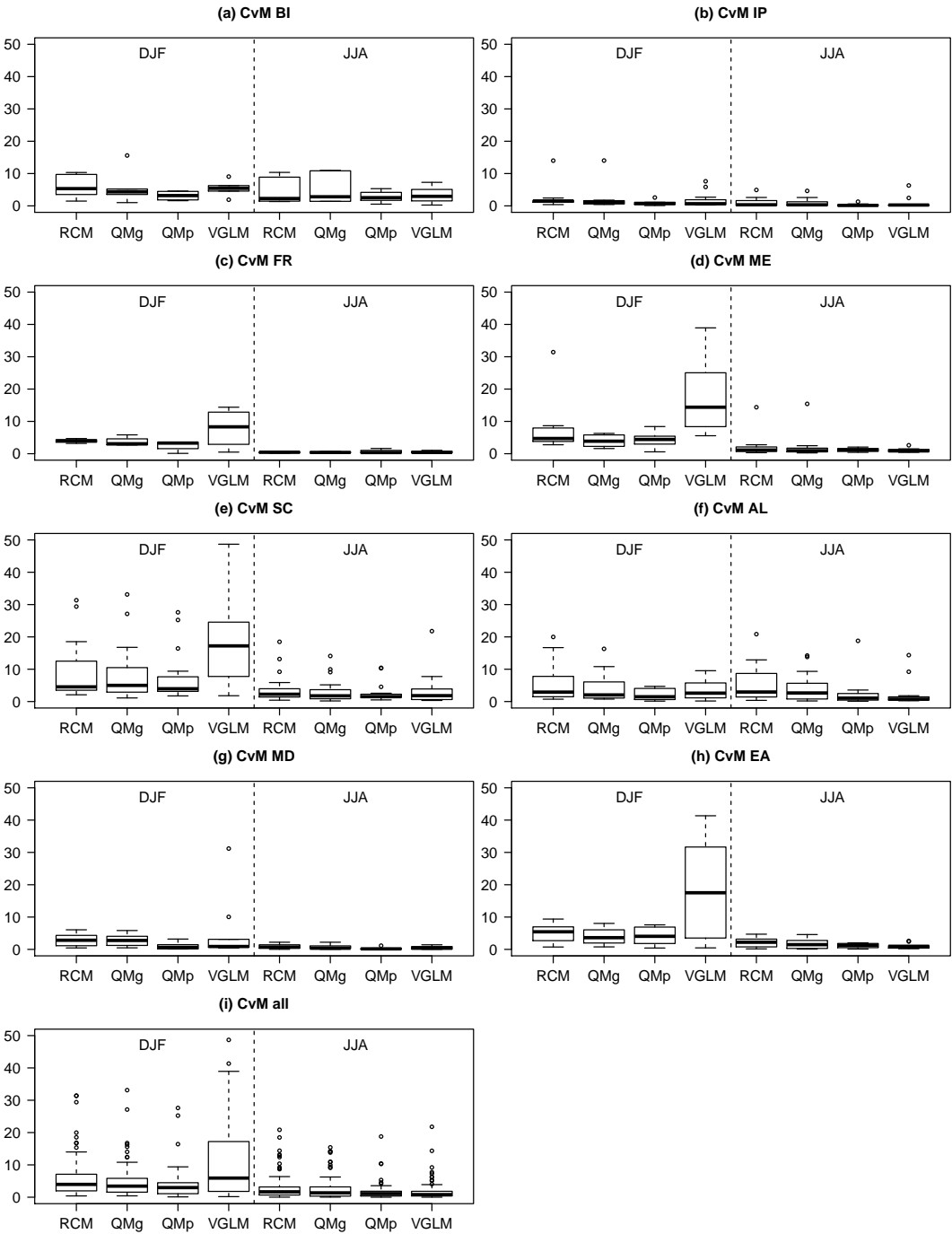

**Figure 11.** Intercomparison of all cross-validated models (not only selected best model). Models: uncorrected RCM, $QM_{grid}$-corrected RCM to grid scale, $QM_{point}$-corrected RCM to point scale and predictor&VGLM-downscaled RCM. Boxplots of CvM-score for all models in different subregions: (a) British Isles (BI), (b) Iberian Peninsula (IP), (c) France (FR), (d) Mid-Europe (ME), (e) Scandinavia (SC), (f) Alps (AL), (g) Mediterranean (MD), (h) eastern Europe (EA) and (i) all locations. For the VGLM the CvM-score was computed for 100 realisations and then averaged. Outlier out of range in (d, i) RCM DJF 54.19.

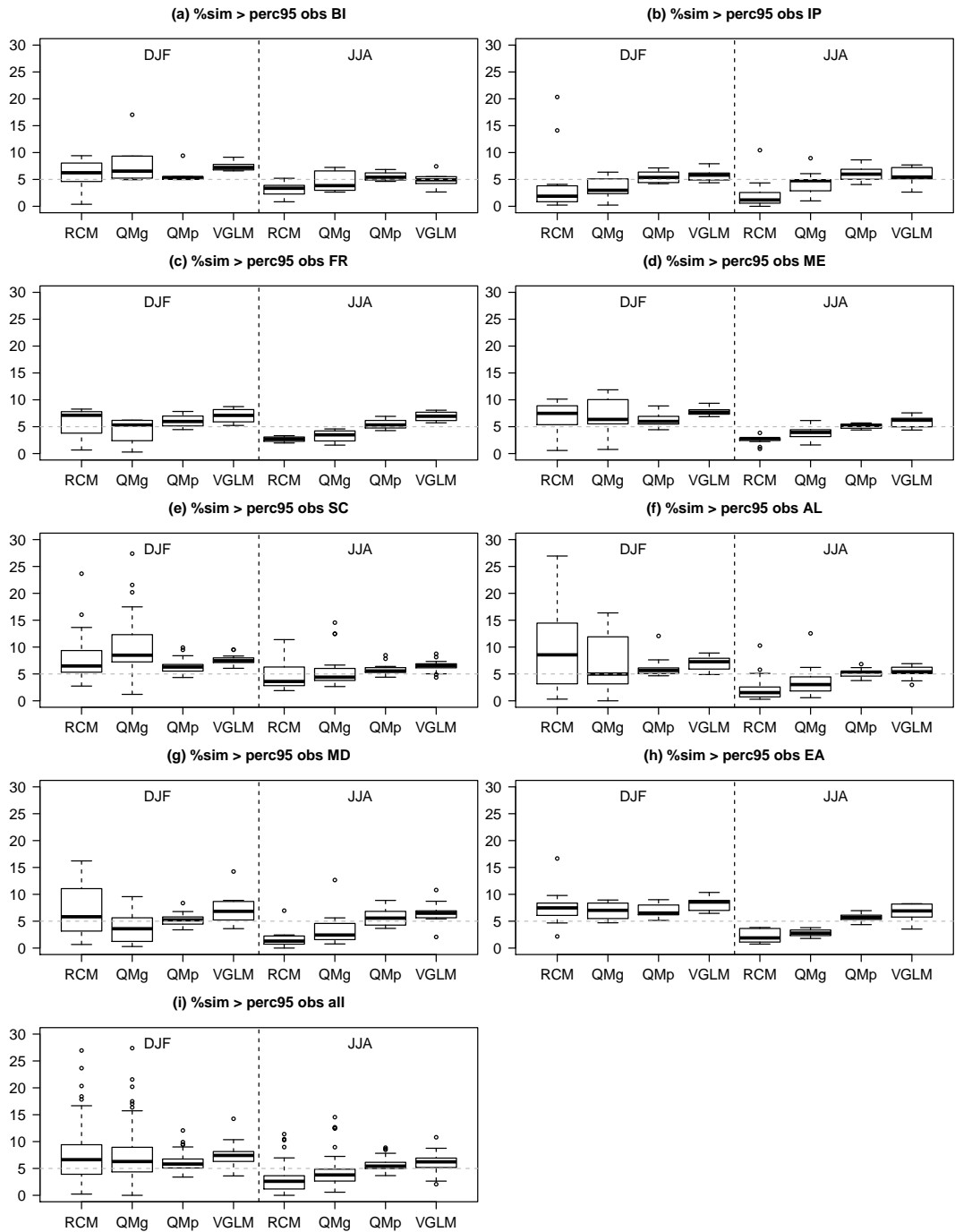

**Figure 12.** As in Fig. 11, but for percentage of simulated wet days exceeding the observed 95th percentile (%sim > perc95$_{obs}$). Outlier out of range in (g, i) QM$_{grid}$ DJF: 41.07 %.

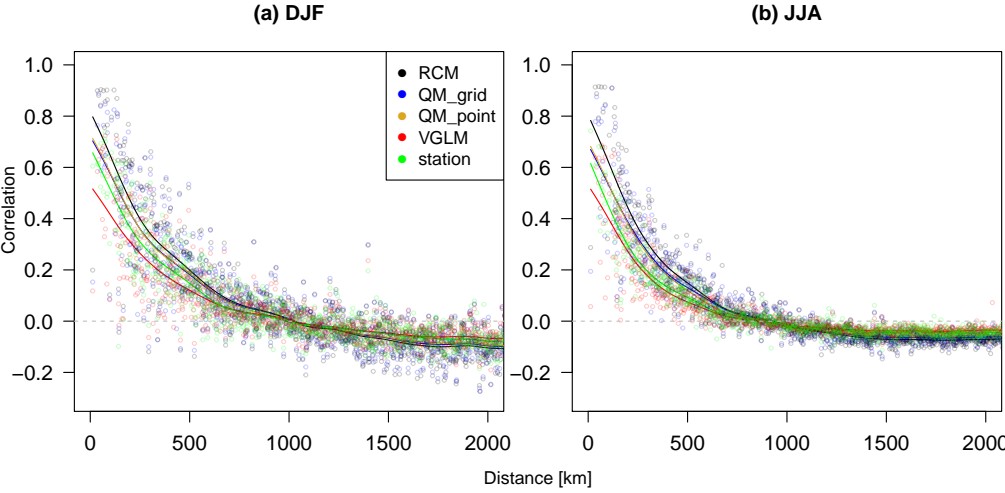

**Figure 13.** Spatial autocorrelation (cross-validated). Correlogram (circles) and smoothed spline fitted to correlogram (lines) for (a) DJF and (b) JJA. Correlogram is estimated by centred Mantel statistic using the R-package ncf (Bjornstad, 2015). For the VGLM 100 realisations of the stochastic model for each station were used to estimate the correlogram.

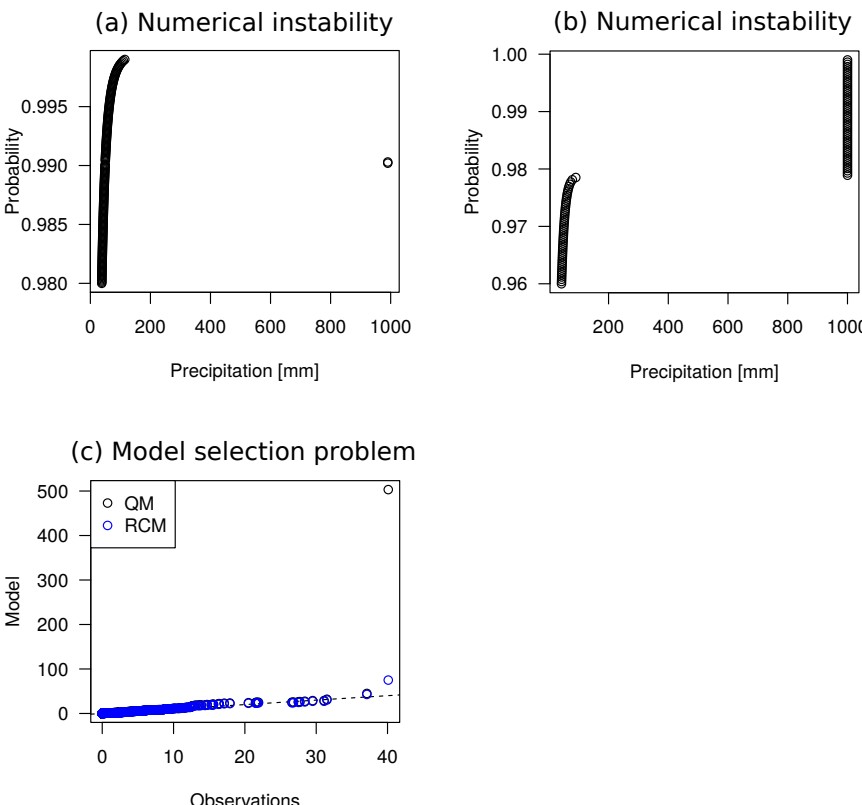

**Figure 14.** Examples for problems with the mixture model. (a) numerical instability: discontinuous cdf, (b) numerical instability: cdf that jumps to the upper bound of 1000 mm d$^{-1}$ and does not jump back as in (a), and (c) problematic model selection: QQ-plot of a selected mixture model that fits well for most quantiles but corrects the extremes to too wet.

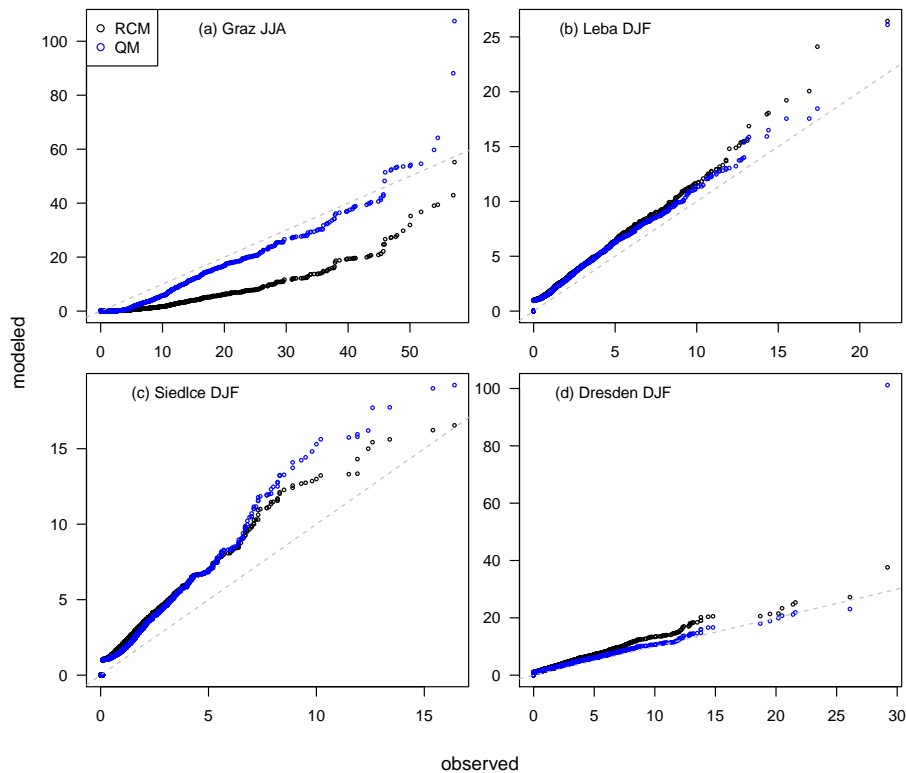

**Figure 15.** Step 1: Bias correction to grid scale. QQ-Plots of RCM-simulated and $QM_{grid}$-corrected (cross-validated) precipitation (mm $d^{-1}$) against E-OBS for stations with high CvM-score. (a) Graz JJA, (b) Leba DJF, (c) Siedlce DJF and (d) Dresden DJF.

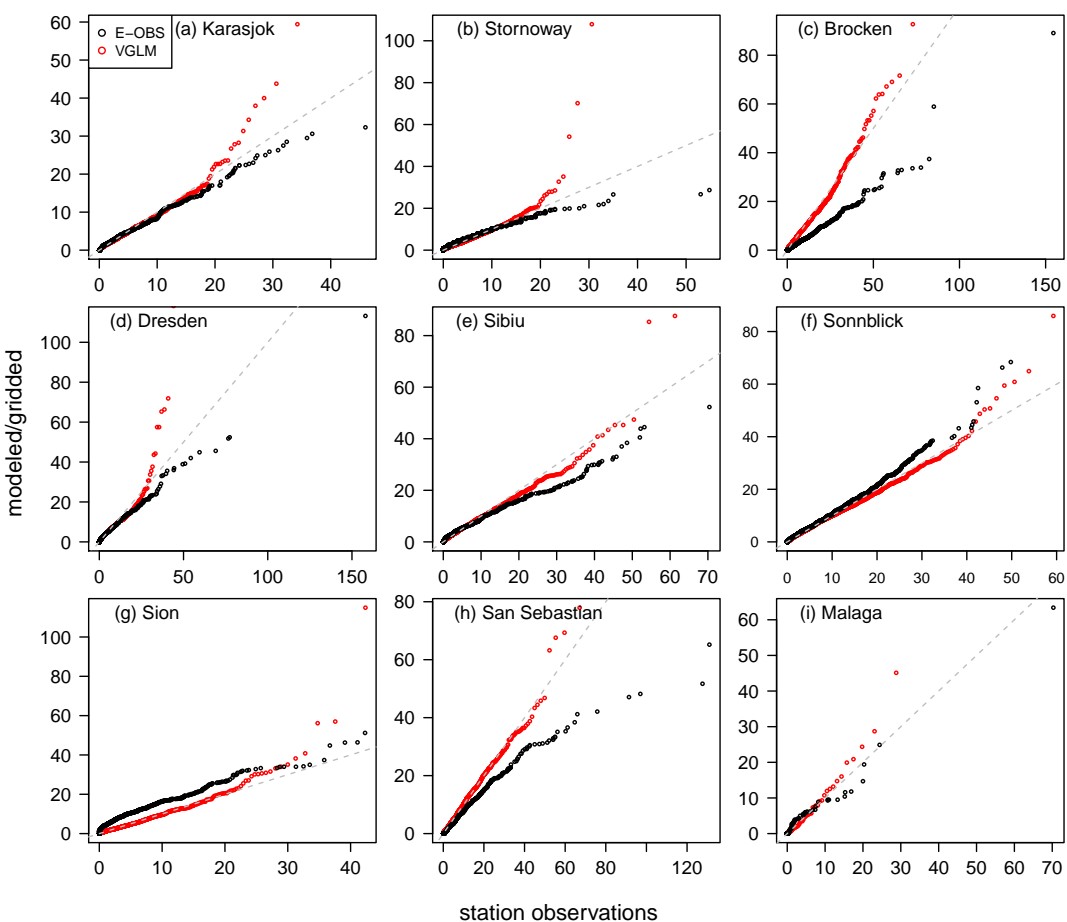

**Figure 16.** Step 2: Downscaling. QQ-Plots for example stations in JJA. VGLM gamma standardized to stationary gamma distribution fitted to observed wet day intensities between gridded and point scale precipitation observations (mm d$^{-1}$). (a) Karasjok, (b) Stornoway, (c) Brocken, (d) Dresden, (e) Sibiu, (f) Sonnblick, (g) Sion, (h) San Sebastian and (i) Malaga.

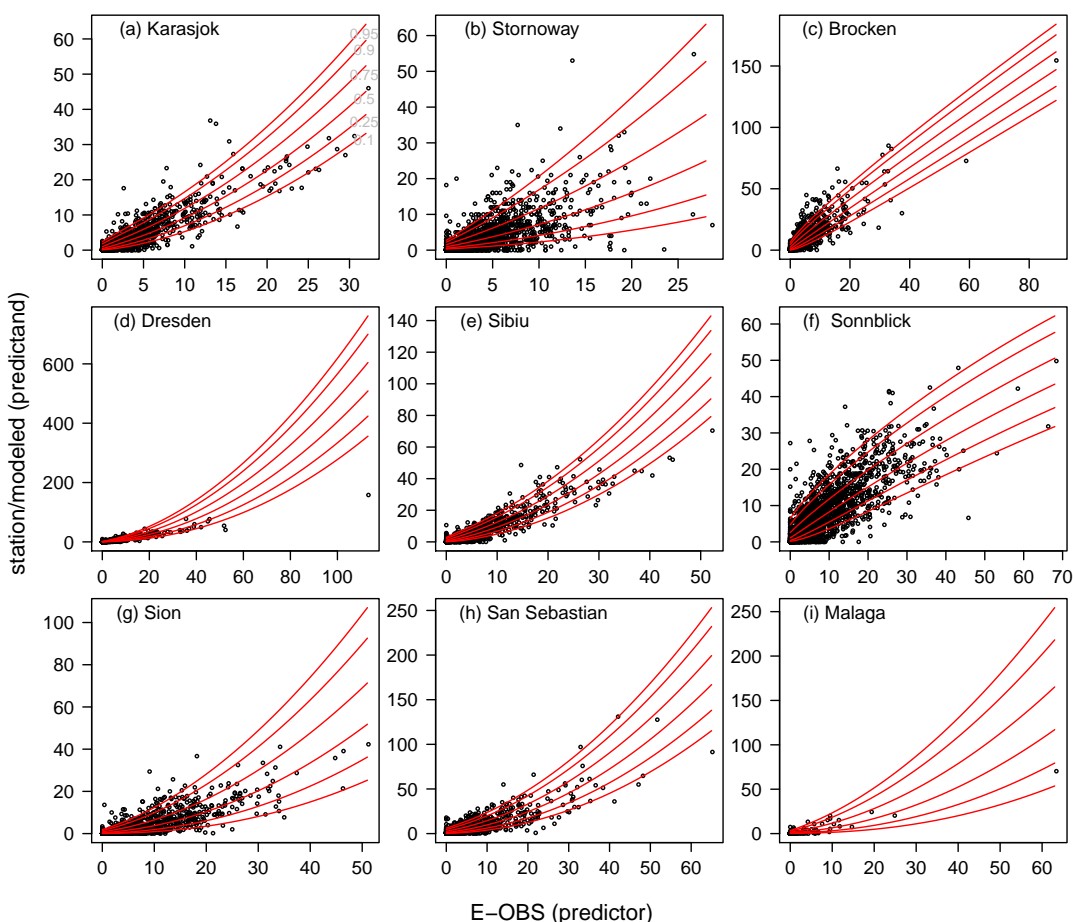

**Figure 17.** Step 2: Downscaling. Estimated relation between gridded and point scale precipitation observations for example stations in JJA. VGLM gamma where both parameters depend on the predictor fitted to observed wet day intensities. The predictor is E-OBS. Circles: observed precipitation intensities (mm d$^{-1}$), lines: 0.1, 0.25, 0.5, 0.75, 0.9 and 0.95 modeled quantile (mm d$^{-1}$). (a) Karasjok, (b) Stornoway, (c) Brocken, (d) Dresden, (e) Sibiu, (f) Sonnblick, (g) Sion, (h) San Sebastian and (i) Malaga.

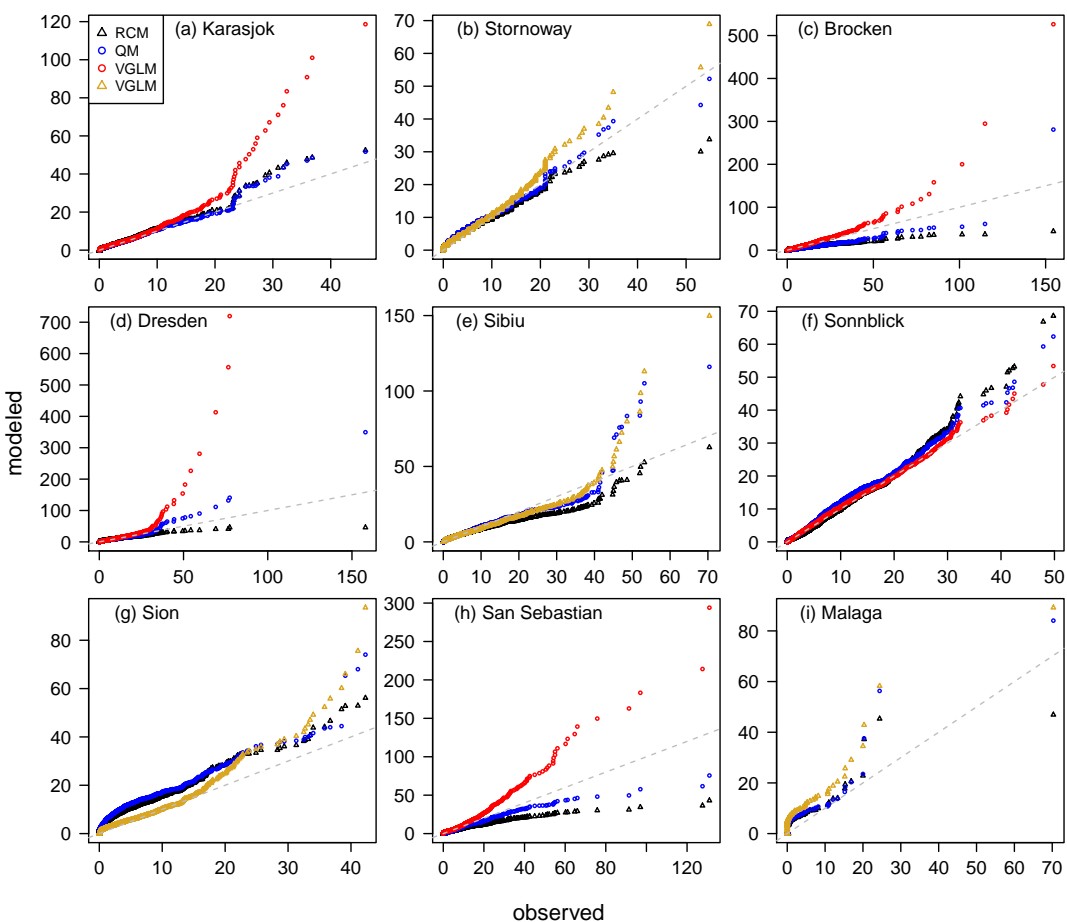

**Figure 18.** QQ-Plots for example stations of different models (cross-validated) against station observations for JJA (mm d$^{-1}$). (a) Karasjok, (b) Stornoway, (c) Brocken, (d) Dresden, (e) Sibiu, (f) Sonnblick, (g) Sion, (h) San Sebastian and (i) Malaga. For the VGLM the quantiles (i.e., sorted time series) of 100 realisations are averaged. Predictor for VGLM as selected by CvM-criterion: (red circles) QM$_{grid}$-bias corrected RCM, (brown triangles) uncorrected RCM. Highest VGLM-modeled quantile in Dresden out of range: 3609 mm d$^{-1}$.