# Peer review of "A Combined Statistical Bias Correction and Stochastic Downscaling Method for Precipitation"

_Hydrology and Earth System Sciences, 2016_

## Referee Comment (RC1) · S. Kotlarski (Referee) · 9 Oct 2016

The work by Volosciuk et al. presents, applies and evaluates a two-step bias correction and downscaling method for daily precipitation. The method consists of the separate application of a parametric quantile mapping (QM) implementation to correct for systematic biases at the spatial scale of the underlying climate model. Second, a vector generalized linear model (VGLM) is applied for stochastic downscaling to the point scale. The approach is applied to a set of 86 stations covering the entire European continent to bias-correct precipitation simulated by a reanalysis-driven regional climate model (RCM). Both steps of the procedure are evaluated separately and also in their combined application. For the latter, the five-fold cross validation framework developed

within the COST Action VALUE is employed. The results indicate an improvement of simulated precipitation characteristics – measured by a CDF score, a score targeting high-intensity events and by the spatial autocorrelation function – by the new method for many cases, but also apparent problems and a deterioration with respect to raw RCM or only bias-corrected RCM output for others.

The manuscript is an excellent piece of work that is highly relevant for the growing climate downscaling community as it introduces a new bias correction and downscaling method that could be employed in a larger frame. The concept to separate bias correction from downscaling is likely to provide advantages compared to simpler approaches, although these advantages are actually not shown (see below). The methods, the underlying data sets and the results are for most parts described appropriately. The quality of presentation is high. The conclusions are – with minor exceptions – properly based on the results obtained. The are no language issues. The work, however, suffers from a number of minor and one major issue that should be improved before final publication of this work. My main concern relates to the missing comparison of the new approach to the widespread "standard" application of QM to both bias-correct and downscale in one step. Please see the listing below for further details. I'd therefore recommend to return the manuscript to the authors for minor revisions. I hope my comments are considered as constructive and will help to improve the (few) weak parts of the paper.

With kind regards, Sven Kotlarski

MAJOR ISSUES

The main motivation of the study relates to the deterministic nature of standard MOS techniques such as QM and their inability to accurately reproduce local-scale variance that is not explained by the actual large-scale predictor. The developed approach is designed to improve on this by separating the bias correction from the downscaling step and by introducing stochasticity into the latter. The performance of both steps

and of the combined scheme is evaluated and compared to the performance of (1) raw RCM output and (2) bias-corrected RCM output (bias correction at the resolution of the RCM). What is missing, however, is a comparison to a "standard" QM application that directly bias-corrects and downscales from the grid cell to the point scale (i.e., the first step of the approach directly targeting the stations series instead of the EOBS grid cell). In my opinion, only such a comparison can show the advantages of the new approach compared to standard applications. This is essential in the light of its apparent problems (at many stations the performance of the two-step procedure is worse than raw RCM or bias-corrected RCM output). I'd therefore suggest to include a fourth dataset in the evaluation of the combined approach, i.e., QM of step 1 directly applied to the station series. This might require to rethink the choice of performance metrics, as I'd expect neither the CvM score nor the 95% score to reveal the advantages of introducing stochasticity. Only the spatial autocorrelation might show such improvements. I'm aware that the suggested extension will to some extent be covered by upcoming VALUE papers, but given the motivation of the two-step approach this comparison is essential for the present paper in my opinion.

MINOR ISSUES (p: page, l: line)

Temporal scale of the model calibration: Evaluation results are presented for both winter (DJF) and summer (JJA), but the temporal scale of the calibration of the two steps remains unclear to me. Have all mdoels be fitted separately for winter and summer, or for the full year, or for every doy-of-the-year with a moving window? This information is a detail, but should be provided.

Extension to further variables: Climate impact studies often require more downscaled variables than precipitation only. Could the presented approach be applied to other variables (e.g., temperature) as well?

Performance measures and model selection: An overview of the performance measures (CvM, 95% score, spatial autocorrelation) is not provided, and most information

has to be taken from the appendix. It would be helpful for the reader to clearly state in Section 2 or 3 which performance measures are applied. Details can still be covered by the appendix. Also, the "model selection" description is a little scattered and hidden. An additional paragraph in the main part of the manuscript clearly outlining the selection strategy (step 1, later on also the best-performing overall approach) would be helpful.

p1 l19: I'd suggest to replace "precipitation data" by "precipitation projections" as this statement primarily concerns future scenario series.

p2 l18: "leading to too smooth variance in space and time" -> I'd suggest to slightly extend this part and to describe the problems of inflation a little more detailed. Readers not familiar with this issue might have problems to understand this limitation. In my opinion, the variance in time (if measured by the SD) should be rather well captured by QM through inflation.

p4 l13: Please check: This 0.44° RACMO simulation might not be the one that is used in the VALUE perfect predictor experiment mentioned before (there it is the corresponding 0.11° simulation to my knowledge). If so, it would be beneficial to briefly mention this fact as it would prevent an inconsistent comparison with the VALUE results.

p4, l27-32: The new method is presented as being versatile and applicable to free-running GCMs/RCMs. While this is true in general, it does not apply to the grid box selection step that relies on temporal correspondence. This should at least briefly be mentioned.

p6 l26: "bias corrected RCM precipitation" (and further occurrences in this section) -> I'd suggest to more generally speak of "coarse-scale precipitation" as not always bias-corrected precipitation is used as predictor (in the evaluation of the second step it is EOBS precipitation, and in the evaluation of the full setup it might also be raw RCM precipitation depending on the selected model).

[Figure]

p8 l5-18: In short, can the remaining inaccuracies of QM be related to non-stationary correction functions (which would show up in the applied cross validation framework)? Section 5.2 (Evaluation of the second step): To evaluate the second step, the modelled precipitation CDFs are used for the QQ plots which requires a standardization to a stationary gamma distribution. Why not using the same framework as for the evaluation of the combined approach, i.e., drawing 100 realizations of precipitation series and then computing the respective percentiles? This would be more straightforward. Also, why is the evaluation of the second step not carried out within the cross validation framework but within a scheme where calibration and validation periods are identical?

p12 l21-34: It remains unclear how the spatial autocorrelation has been computed. Based on seasonal means? Or separately for each day and then averaged?

p14 l14-16: This statement is a little overconfident given the results previously shown. In many regions VGLM is comparable, sometimes even inferior to raw or bias-corrected RCM (as said above, an extra comparison to QM directly onto the station series would be very beneficial here). Concerning the spatial autocorrelation, at least in DJF I wouldn't speak of an improvement by VGLM.

p14 l16-18: This sentence requires a proper reference (this aspect is not covered by the work present).

User guidance: Given the remaining apparent problems of the new two-step approach at many sites (inferiority even against raw RCM data), some summarizing guidance would be helpful on when to use the new scheme and when this is considered critical. This guidance should account for the fact, that in a free-running setup which is the setup for climate scenarios, not all parts of the presented evaluation can be carried out beforehand (no temporal correspondence which prevents the use of the same strategy for grid box selection and to some extent also the cross-validation setup with short time slices).

---

## Referee Comment (RC2) · S. Hagemann (Referee) · 14 Oct 2016

**Manuscript:** A Combined Statistical Bias Correction and Stochastic Downscaling Method for Precipitation

**Major remarks**

The authors present a bias correction approach for GCM or RCM precipitation where this post processing is separated into two steps. Step 1 comprises the pure bias correction using a quantile-mapping method at the same scale as the RCM data that are corrected in the study. In Step 2, a downscaling method is applied from the grid scale to station locations. Using vector generalized linear gamma model (VGLM), downscaling parameters are derived from gridded and station observations, and then these calibrated VGLM is applied to the RCM data. This separation into bias correction and downscaling is an innovative and promising approach and, hence, of interest for a wider scientific community.

There are a few points that should be improved before publication:

The selection procedure and evaluation in step 1 is not well described, e.g. on p.7 = Sect. 3.1. Usually, I would expect that only bias-corrected data are used in step 2, which is not the case. The selection procedure is well described in the appendix A2 (especially lines 37-40 on p. 16) but not in the main text. Also I would expect that a bias corrected precipitation map is compared with the uncorrected precipitation data and observations. But suddenly a predictor is mentioned instead of precipitation, and a mixed map is shown only for the station locations. This is rather confusing when first reading the paper.

In step 1, the quantile mapping bias correction improves RCM precipitation in 73 of 86 cases in DJF, but only for 49 of 86 cases in JJA. As quantile mapping can be a rather powerful approach, it seems that the chosen gamma function for the transfer functions fails in a lot of cases especially in JJA. Concluding from this it may be suitable to point to approaches (in the discussion section) where several functions can be used as candidate for the transfer function, such it has been done, e.g. by (Piani, C., G.P. Weedon, M. Best, S.M. Gomes, P. Viterbo, S. Hagemann, and J. O. Haerter, 2010: Statistical bias correction of global simulated daily precipitation and temperature for the application of hydrological models. J. Hydrol., 395, doi:10.1016/j.jhydrol.2010.10.024, 199–215)

For step 2, it seems even worse. Here, the combined approach provides the best precipitation estimate only for 25 (45) of 86 stations in DJF (JJA). Thus on a first glance, the application of the chosen VLGM does not appear to be a suitable method for the downscaling that can be recommended. What would happen if you use the quantile mapping to directly bias correct the RCM data to the station observations, i.e. using an approach such as it commonly done in bias correction literature? How this would compare to your results?

Actually, I really like the approach of separating bias correction and downscaling. But the rather poor results of the downscaling step may obscure the innovation of the used approach. Thus, some of my remarks aim at that this obscuration does not happen. In summary I suggest accepting the paper for publication after some revisions have been conducted.

I don't wish do stay anonymous, Stefan Hagemann

**Minor remarks**

In the following suggestions for editorial corrections are marked in *Italic*.

p.4 – line 1
Please provide an explanation for readers that are not familiar with the "five-fold cross validation". Either short in the main text or long in the appendix with a reference to this in the main text.

p.4 – line 19-20
Although E-OBS is probably *not an appropriate reference in some regions* it …

p.6 – line 28
The term "logit link function" is not common knowledge. Please explain!

p.7 – line 26
…version *since the* calibration …

p.8 – line 31
… performance *of the VGLM gamma for different climates*, …

p.9 – line 5-7
It is written:
To evaluate the goodness of fit we use residual QQ-plots (Fig. 6 for DJF and Fig. 8 for JJA). As a QQ-plot requires quantiles of an unconditional distribution we standardized the from day-to-day varying distribution to a stationary gamma distribution (Coles, 2001; Wong et al., 2014). Thereby the effect of the predictor is approximately removed.

What do you mean with "the effect of the predictor is approximately removed"? You are using E-OBS as predictor. If the effect of E-OBS is removed, would you get the same results with any other predictor? I don't understand.

p.10 – line 10
…correction, *section* 5.1) …

p.11 – line 29
This *raises* the question …

p.12 – line 30-32
It is written: " The spatial …improved by the stochastic downscaling step."

Obviously this statement is correct for DJF. If half of the regions get worse with the downscaling, this questions the general usefulness of the chosen VGLM method (see also major remarks).
In DJF, precipitation is generally strongly determined by the large-scale circulation. Here, the QM bias correction already yields quite good corrected precipitation values. But why the VGLM makes it worse in the majority of cases? This implies a strong weakness of the chosen downscaling method. You provide some potential reasons, but I suggest also coming up with some more details on how this may be improved. You may even undermine this with examples for single stations. For example, you mention "For instance, another distribution in the VGLM…." on p. 15 – line 3-5. Is it possible to provide a plot for one station where another distribution/function is used that improves the downscaling for this particular station?

Figure 2
I suggest including the PRUDENCE regions in the plot as a major part of your evaluation is based on these regions. For example increase the size of the figure and include PRUDENCE regions as boxes with another colour, e.g. red.

Figure 3 caption
… QM *corrected RCM*, triangles: *uncorrected* RCM.

Figure 7 and 9
What benefit do Fig. 7 and 9 provide? Are they necessary or can they be removed?

---

## Referee Comment (RC3) · Anonymous Referee #3 · 17 Oct 2016

The authors present a new approach to bias correct and downscale precipitation from free running GCM/RCM simulations. The authors use the ERA interim nested KNMI RACMO2 RCM, the gridded E-OBS field and E-OBS station data to test their method. The evaluation is focussing on each modelling step (i.e., bias correction and down-scaling separately as well as both combined). Overall, the evaluation is very compre-hensive and the study is a welcome contribution to the highly relevant field of climate impact analysis. Also the motivation and description of the method is nicely described and easy to understand. However, the manuscript is imbalanced with respect to usage of figures and text in the results section, with too little text for too many figures. Also, the conclusions are too a large extent a summary of the results and provide only little

discussion (I only found p.15 l. 2-5) and conclusion. The result of this is that the paper reads very much like a technical report. The authors should much more discussions on the results with references in the results part. This discussion should include the following points.

Major Comments

The presented method of VGLM + QM corrected RCM is not able to outperform the low resolution data (i.e. the raw RCM data or the QM corrected data) everywhere. See for example Figure 10, where the RCM (triangle) and QM (circles) is present for most locations. How can it be that a statistical post-processing is decreasing the performance? It should at least be as good as the gridded data. This indicates to me that precipitation is to a large extend not following a gamma distribution as used in the VGLM and that the linear predictor-predictand relationship (eq. 6) is implausible. The authors should provide an explanation on this point and also relate their findings to that of previous studies such at Wong et al. 2014 or Eden et al. 2014.

Additionally, there are some contradictions that must be resolved. For example, the authors state on p. 14 l. 26ff, that E-OBS might be unreliable in France and eastern Europe due to low station density that implies a misrepresentation of gridded precipitation in this region. But on p. 12 l. 4, that in Scandinavia E-OBS has a high station density and is of good quality, but the VGLM is still performing poorly. This indicates to me that the quality of E-OBS cannot be identified as the source of bad performance for the VGLM.

The results for the different European regions (e.g., Figure 5) should also be related to previous research as tremendous research has used this classification.

Minor comment

The authors should include an appendix shortly summarising the approach by Wong et al. or include this in the methods part as it is important for the reader to understand

the difference between the method by Wong et al. and the one presented here.

The ordering of Figures should follow the order they are referred to in the text.

p. 4, l.14ff: "We have...", I do not understand this sentence. Please rephrase.

p. 6. l.4f: Assuming that precipitation is in every case heavy tailed seems like a strong assumption. Could this assumption not also lead to overestimation of extremes as seen for the VGLM in Figure 12?

p.7 l.28: The author should give more details for the cross-validation setup. I assume it is in time, but I am not sure which periods have been used for calibration/validation.

p.11 l. 6: I think that Figure 10d) shows that the model is strongly underestimating the occurence of heavy precipitation events by almost 50% in most locations.

p.12 l.21ff: How is the correlogram calculated for the 100 VGLM realizations. Are the realizations first averaged and is the correlation calculated afterwards or the other way around? Please add this also in the text.

p.13 l.9ff: The improvement of the "drizzle effect", "location bias" has not been shown in this study but in previous work. The references should be added to avoid misunderstanding.
* * *

---

## Author Comment (AC1) · 9 Dec 2016

We would like to thank the reviewers for carefully reading our manuscript and for their constructive comments which will improve our manuscript. We agree with most suggestions and we will implement them in a revised version.

As major change we will add a comparison of our results to the quantile mapping (QM) approach applied between RCM and point scale (i.e., station). In Fig. 1 the QM between RCM and point scale (i.e., station) has been added to the spatial autocorrelation plot Fig. 16 of the manuscript. The spatial autocorrelation of the QM between RCM and point scale is very similar to that of the QM between RCM and grid scale. This confirms that the QM-approach is not capable to model small scale variability, and that a stochastic model is needed to bridge the scale gap. Particularly in summer (JJA) the VGLM improves spatial autocorrelation compared to QM for both RCM to grid and RCM to point scale.

[Figure]

*Fig.1: Spatial autocorrelation (cross-validated). Correlogram (circles) and smoothed spline fitted to correlogram (lines). Correlogram is estimated by centred Mantle statistic using the R-package ncf (Bjornstad, 2015). For the VGLM 100 realisations of the stochastic model for each station were used to estimate the correlogram. Difference to Fig. 16 from the manuscript: quantile mapping between RCM and point scale (station) added.*

Please find a detailed response below.

**Response to Reviewer #1**

MAJOR ISSUES

> *The main motivation of the study relates to the deterministic nature of standard MOS techniques such as QM and their inability to accurately reproduce local-scale variance that is not explained by the actual large-scale predictor. The developed approach is designed to improve on this by separating the bias correction from the downscaling step and by introducing stochasticity into the latter. The performance of both steps and of the combined scheme is evaluated and compared to the performance of (1) raw RCM output and (2) bias-corrected RCM output (bias correction at the resolution of the RCM). What is missing, however, is a comparison to a "standard" QM application that directly bias-corrects and downscales from the grid cell to the point scale (i.e., the first step of the approach directly targeting the stations series instead of the EOBS grid cell). In my opinion, only such a comparison can show the advantages of the new approach compared to standard applications. This is essential in the light of its apparent problems (at many stations the performance of the two-step procedure is worse than raw RCM or bias-corrected RCM output). I'd therefore suggest to include a fourth dataset in the evaluation of the combined approach, i.e., QM of step 1 directly applied to the station series. This might require to rethink the choice of performance metrics, as I'd expect neither the CvM score nor the 95% score to reveal the advantages of introducing stochasticity. Only the spatial autocorrelation might show such improvements. I'm aware that the suggested extension will to some extent be covered by upcoming VALUE papers, but given the motivation of the two-step approach this comparison is essential for the present paper in my opinion.*

We agree with the reviewer that comparing our results to the standard quantile mapping approach from RCM to station data would improve our manuscript and we will add this comparison in a revised version (see above).

MINOR ISSUES (p: page, l: line)

> *Temporal scale of the model calibration: Evaluation results are presented for both winter (DJF) and summer (JJA), but the temporal scale of the calibration of the two steps remains unclear to me. Have all mdoels be fitted separately for winter and summer, or for the full year, or for every doy-of-the-year with a moving window? This information is a detail, but should be provided.*

The model has been calibrated for each season separately. This information will be added to a revised version of the manuscript.

*Extension to further variables: Climate impact studies often require more downscaled variables than precipitation only. Could the presented approach be applied to other variables (e.g., temperature) as well?*

In principle this approach should be applicable to any variable that is gamma distributed. For, e.g., temperature a model based on a normal distribution might be more suitable. Nevertheless, at this time our approach has only been evaluated for precipitation. Transferring it to other variables would require an evaluation for the particular variable. We will discuss this issue in the revised manuscript.

*Performance measures and model selection: An overview of the performance measures (CvM, 95% score, spatial autocorrelation) is not provided, and most information has to be taken from the appendix. It would be helpful for the reader to clearly state in Section 2 or 3 which performance measures are applied. Details can still be covered by the appendix. Also, the "model selection" description is a little scattered and hidden. An additional paragraph in the main part of the manuscript clearly outlining the selection strategy (step 1, later on also the best-performing overall approach) would be helpful.*

In a revised version of the manuscript we will add an overview of the applied performance measures and add a description of the model selection procedures to the main part of the manuscript.

*p1 l19: I'd suggest to replace "precipitation data" by "precipitation projections" as this statement primarily concerns future scenario series.*

Will be replaced.

*p2 l18: "leading to too smooth variance in space and time" -> I'd suggest to slightly extend this part and to describe the problems of inflation a little more detailed. Readers not familiar with this issue might have problems to understand this limitation. In my opinion, the variance in time (if measured by the SD) should be rather well captured by QM through inflation.*

We are happy to extend the discussion of this issue. Anyway, also the temporal variability is in general affected by inflation (see, e.g., von Storch, J Climate, 1999). Local precipitation is a randomly disaggregated grid-box precipitation, or vice versa, grid box precipitation is the area average of sub-grid precipitation. The temporal memory is then higher at the grid-box scale than at the local scale, i.e., grid-box time series are smoother in time than local series. Quantile mapping cannot overcome this mismatch in temporal structure (apart from correcting the drizzle effect). This effect is, of course, less pronounced than the spatial effect as demonstrated in Maraun, J. Climate, 2013. Actually, the results of the COST Action VALUE (currently submitted) indicate this effect.

*p4 l13: Please check: This 0.44◦ RACMO simulation might not be the one that is used in the VALUE perfect predictor experiment mentioned before*

*(there it is the corresponding 0.11◦ simulation to my knowledge). If so, it would be beneficial to briefly mention this fact as it would prevent an inconsistent comparison with the VALUE results.*

The reviewer is right that the RACMO simulation we used differs in resolution from the one used in VALUE. In a revised version of the manuscript we will mention this difference.

*p4, l27-32: The new method is presented as being versatile and applicable to freerunning GCMs/RCMs. While this is true in general, it does not apply to the grid box selection step that relies on temporal correspondence. This should at least briefly be mentioned.*

The reviewer is right that the grid box selection step must be calibrated with a reanalysis-driven RCM-simulation to ensure temporal correspondance. As the location bias is specific to the employed RCM, and most likely not dependent on the driving model, the selected grid box as calibrated by a reanalysis-driven simulation of the same RCM can be applied to the free-running simulation. However, the impact of the driving model on the grid box that best represents local climate should be checked in future work. We will clarify this in a revised version of our manuscript.

*p6 l26: "bias corrected RCM precipitation" (and further occurrences in this section) -> I'd suggest to more generally speak of "coarse-scale precipitation" as not always biascorrected precipitation is used as predictor (in the evaluation of the second step it is EOBS precipitation, and in the evaluation of the full setup it might also be raw RCM precipitation depending on the selected model).*

We agree that the wording suggested by the reviewer is more consistent and we will change this in a revised version.

*p8 l5-18: In short, can the remaining inaccuracies of QM be related to non-stationary correction functions (which would show up in the applied cross validation framework)?*

The remaining inaccuracies of QM can be related to both a time-varying correction function as suggested by the reviewer and the parametric correction function. We will mention this in a revised version.

*Section 5.2 (Evaluation of the second step): To evaluate the second step, the modelled precipitation CDFs are used for the QQ plots which requires a standardization to a stationary gamma distribution. Why not using the same framework as for the evaluation of the combined approach, i.e., drawing 100 realizations of precipitation series and then computing the respective percentiles? This would be more straightforward. Also, why is the evaluation of the second step not carried out within the cross validation framework but within a scheme where calibration and validation periods are identical?*

For the second step we show standardized QQ-Plots where the effect of the predictor is approximately removed. This allows to evaluate the goodness of fit of the model which is our aim here. Drawing 100 realisations would include both the fitted model and the effect of the predictor. This comment of the reviewer clearly shows that we did not explain this sufficiently in the current version of our manuscript. In a revised version we will clarify this.

The problems with the chosen model in some locations are already present when repredicting the calibration period where the skill should be higher than in a cross validation where a period is predicted that is not part of the calibration period. This clearly highlights deficiencies in the model for these locations even in this setup where the skill should be higher. We will mention this in a revised version of the manuscript.

> *p12 l21-34: It remains unclear how the spatial autocorrelation has been computed. Based on seasonal means? Or separately for each day and then averaged?*

The spatial autocorrelation is computed based on daily values and then averaged. We will add an explanation to the revised manuscript about the details of the spatial autocorrelation.

> *p14 l14-16: This statement is a little overconfident given the results previously shown. In many regions VGLM is comparable, sometimes even inferior to raw or bias-corrected RCM (as said above, an extra comparison to QM directly onto the station series would be very beneficial here). Concerning the spatial autocorrelation, at least in DJF I wouldn't speak of an improvement by VGLM.*

In a revised version this part will be reformulated. Here we rather show a concept than the perfect model for each station which we will highlight in a revised manuscript. Further research on the specific implementation is required for some regions. The reviewer is right that in DJF the noise component is slightly too strong in our model. This could be improved by a multi-site model which is beyond the scope of our study. In DJF a stochastic model is generally less important as more variability is explained by the grid box than in JJA where precipitation is often caused by small scale convective events. We agree that the extra comparison of the QM between RCM and point scale is beneficial here. As explained above this will be added in a revised version of the manscript.

> *p14 l16-18: This sentence requires a proper reference (this aspect is not covered by the work present).*

This aspect is a simple consequence of using a regression model instead of inflation – in QM, the inflation is responsible for changes in trends. We will cite Maraun, J Climate, 2014, where this issue is explained.

> *User guidance: Given the remaining apparent problems of the new two-step approach at many sites (inferiority even against raw RCM data),*

*some summarizing guidance would be helpful on when to use the new scheme and when this is considered critical. This guidance should account for the fact, that in a free-running setup which is the setup for climate scenarios, not all parts of the presented evaluation can be carried out beforehand (no temporal correspondence which prevents the use of the same strategy for grid box selection and to some extent also the cross-validation setup with short time slices).*

We will add a paragraph to guide users in a revised version. In this paragraph we would highlight that for some regions a specific implementation different from the one we used is required. We will also highlight that this work rather introduces a concept than providing the perfect specific implementation for each site. Moreover, our evaluation shows that bias correction and downscaling methods must not be capriciously transferred from one region to another. Users need to re-evaluate the method when transferring it to locations with different climatic conditions. The concept, however, can be extended to a wide range of method combinations.

**Response to Reviewer #2**

Major remarks

> *The selection procedure and evaluation in step 1 is not well described, e.g. on p.7 = Sect. 3.1. Usually, I would expect that only bias-corrected data are used in step 2, which is not the case. The selection procedure is well described in the appendix A2 (especially lines 37-40 on p. 16) but not in the main text. Also I would expect that a bias corrected precipitation map is compared with the uncorrected precipitation data and observations. But suddenly a predictor is mentioned instead of precipitation, and a mixed map is shown only for the station locations. This is rather confusing when first reading the paper.*

As already mentioned in the response to a similar comment by Reviewer #1 the description of the model selection procedure and an overview of the evaluation metrics will be provided in the main manuscript in a revised version.

We agree with the reviewer that it would be easier for the reader if the results part is started with a more general representation of our results. Thus, in a revised version we will start the results part with showing maps of the mean bias of the raw uncorrected RCM, the standard quantile mapping between RCM and station, and our method.

> *In step 1, the quantile mapping bias correction improves RCM precipitation in 73 of 86 cases in DJF, but only for 49 of 86 cases in JJA. As quantile mapping can be a rather powerful approach, it seems that the chosen gamma function for the transfer functions fails in a lot of cases especially in JJA. Concluding from this it may be suitable to point to approaches (in the discussion section) where several functions can be used as candidate for the transfer function, such it has been done, e.g. by (Piani, C., G.P. Weedon, M. Best, S.M. Gomes, P. Viterbo, S. Hagemann, and J. O. Haerter, 2010: Statistical bias correction of global simulated daily precipitation and temperature for the application of hydrological models. J. Hydrol., 395, doi:10.1016/j.jhydrol.2010.10.024, 199–215)*

In a revised version we will mention the suggested paper in the discussion section. Another reason why the QM bias correction improves precipitation in less cases in JJA could be that the bias in the RCM is lower in JJA than in DJF (see Fig. 14 and also Kotlarski et al., 2014).

> *For step 2, it seems even worse. Here, the combined approach provides the best precipitation estimate only for 25 (45) of 86 stations in DJF (JJA). Thus on a first glance, the application of the chosen VLGM does not appear to be a suitable method for the downscaling that can be recommended. What would happen if you use the quantile mapping to directly bias correct the RCM data to the station observations, i.e. using an approach such as it commonly done in bias correction literature? How this would compare to your results?*

As also suggsted by Reviewer #1 and mentioned above a comparison of our results to the QM applied between RCM and point scale (i.e., station) will be added to a revised version of our manuscript. The spatial autocorrelation of the QM between RCM and point scale is very similar to that of the QM between RCM and grid scale (see Fig. 1). This confirms that the QM-approach is not capable to model small scale variability, and that a stochastic model is needed to bridge the scale gap. Yet, the reviewer is right that the specific implementation that we employed here is not suitable for all locations. In the revised manuscript we will point out that we rather introduce a concept than the perfect specific model for all locations which should be subject of future studies however.

Minor remarks

> *p.4 – line 1 Please provide an explanation for readers that are not familiar with the "five-fold cross validation". Either short in the main text or long in the appendix with a reference to this in the main text.*

Will be provided.

> *p.4 – line 19-20 Although E-OBS is probably* not an appropriate reference in some regions *it ...*

Will be corrected.

> *p.6 – line 28 The term "logit link function" is not common knowledge. Please explain!*

An explanation will be added.

> *p.7 – line 26 ...version* since the *calibration ...*

Will be corrected.

> *p.8 – line 31 ... performance* of the VGLM gamma for different climates, ...

Will be corrected.

> *p.9 – line 5-7 It is written: To evaluate the goodness of fit we use residual QQ-plots (Fig. 6 for DJF and Fig. 8 for JJA). As a QQ-plot requires quantiles of an unconditional distribution we standardized the from day-to-day varying distribution to a stationary gamma distribution (Coles, 2001; Wong et al., 2014). Thereby the effect of the predictor is approximately removed.*
> *What do you mean with "the effect of the predictor is approximately removed"? You are using E-OBS as predictor. If the effect of E-OBS is removed, would you get the same results with any other predictor? I don't understand.*

The parameters of the gamma distribution for a particular day are determined by the estimated regression parameters of the VGLM and the predictor (see Eq.

6 in the manuscript). As the gamma-parameters depend on the predictor they vary from day-to-day due to the varying predictor. The applied standardization transforms these from day-to-day varying gamma-parameters to a stationary gamma distribution. This stationary distribution has no longer the predictor dependent day-to-day variations. Or in other words the effect of the predictor is approximately removed. Due to this procedure the goodness of fit of the regression model can be evaluated instead of evaluating the combined effect of predictor and regression model which is usually present in the from day-to-day varying gamma-parameters. These results should be very similar with any other predictor. Therefore, deficiencies that are indicated by these results are either due to inappropriate model structure or not well fitting parameters which could be caused by an inappropriate calibration dataset.

This comment of the reviewer shows that we did not explain this procedure sufficiently to make it easily understandable for the reader. In a revised version of the manuscript we will add a more detailed explanation on this.

> *p.10 – line 10 …correction,* section *5.1) …*

Will be changed.

> *p.11 – line 29 This* raises *the question …*

Typo error will be corrected.

> *p.12 – line 30-32 It is written: " The spatial …improved by the stochastic downscaling step." Obviously this statement is correct for DJF. If half of the regions get worse with the downscaling, this questions the general usefulness of the chosen VGLM method (see also major remarks). In DJF, precipitation is generally strongly determined by the large-scale circulation. Here, the QM bias correction already yields quite good corrected precipitation values. But why the VGLM makes it worse in the majority of cases? This implies a strong weakness of the chosen downscaling method. You provide some potential reasons, but I suggest also coming up with some more details on how this may be improved. You may even undermine this with examples for single stations. For example, you mention "For instance, another distribution in the VGLM…." on p. 15 – line 3-5. Is it possible to provide a plot for one station where another distribution/function is used that improves the downscaling for this particular station?*

The reviewer is right that further research is required to find the appropriate stochastic model for each of the locations. However, this is beyond the scope of our study. As the applied correction function appears to be not flexible enough in some regions we have tried to implement a VGLM with splines. This implementation is unfortunately not straightforward and comes along with the risk of overfitting. We agree that this issue should be implemented and carefully evaluated in a follow-up study. In a revised version we will highlight that the aim of this study is to introduce the concept of combining a bias

correction with a downscaling method rather than finding the perfect specific implementation for each location.

> *Figure 2 I suggest including the PRUDENCE regions in the plot as a major part of your evaluation is based on these regions. For example increase the size of the figure and include PRUDENCE regions as boxes with another colour, e.g. red.*

This suggestion of the reviewer is a very good idea and will be added in a revised version of the manuscript.

> *Figure 3 caption … QM corrected RCM, triangles: uncorrected RCM.*

Will be changed.

> *Figure 7 and 9 What benefit do Fig. 7 and 9 provide? Are they necessary or can they be removed?*

The circles in these figures show which predictands there are on the point scale for a given grid scale predictor. The lines show different quantiles of the modeled distribution for the given predictor. On the one hand this shows how well the model fits the observational relationship. While this information is partly redundant to the QQ-plots these plots also provide evidence for potential explanations why the relationship is in some locations not well. The VGLM currently allows basically for three different model behaviors: concave (i.e., Brocken DJF), straight (i.e., San Sebastian DJF) or convex (i.e., Malaga DJF). This appears to be not flexible enough for some locations. No changes between these three types are possible. A more flexible relationship that allows for a changed model behavior for higher values could improve the results but comes along with the risk of overfitting.

More discussion on these figures will be added in a revised version.

**Response to Reviewer #3**

Major Comments

> *The presented method of VGLM + QM corrected RCM is not able to outperform the low resolution data (i.e. the raw RCM data or the QM corrected data) everywhere. See for example Figure 10, where the RCM (triangle) and QM (circles) is present for most locations. How can it be that a statistical post-processing is decreasing the performance? It should at least be as good as the gridded data. This indicates to me that precipitation is to a large extend not following a gamma distribution as used in the VGLM and that the linear predictor-predictand relationship (eq. 6) is implausible. The authors should provide an explanation on this point and also relate their findings to that of previous studies such at Wong et al. 2014 or Eden et al. 2014.*

We discuss potential reasons for the problems with the VGLM including an implausible linear relationship in the manuscript. As the reviewer we were also concerned that precipitation might not be gamma-distributed in these regins. Therefore, we verified that precipitation is gamma distributed in these locations. This comment of the reviewer shows that although not shown in the manuscript this shall be mentioned in a revised version to avoid confusion.

The studies by Wong et al. and Eden et al. cover the British Isles, and thus, do not tackle the climates where we find problems, i.e., continental winter climates. This shows that bias correction and downscaling methods must not be transferred from one region to another without reevaluation for the climatic conditions of the region where it is transfered to. We will discuss these issues in a revised version.

> *Additionally, there are some contradictions that must be resolved. For example, the authors state on p. 14 l. 26ff, that E-OBS might be unreliable in France and eastern Europe due to low station density that implies a misrepresentation of gridded precipitation in this region. But on p. 12 l. 4, that in Scandinavia E-OBS has a high station density and is of good quality, but the VGLM is still performing poorly. This indicates to me that the quality of E-OBS cannot be identified as the source of bad performance for the VGLM.*

We agree that the quality of E-OBS can not be identified as the only source for bad performance. Nevertheless, we think that it is a potential reason amongst others that could be adressed and quantified in a future study. We will clarify this in a revised version.

> *The results for the different European regions (e.g., Figure 5) should also be related to previous research as tremendous research has used this classification.*

The results for the different European regions will be compared to Dosio and Paruolo, 2011 (JGR). In agreement with our results they get large improvements

over the Alps, Spain and France. Yet, in contrast to our results they also get good results for middle and eastern Europe where we find persisting biases even after bias correction.

Minor comment

> *The authors should include an appendix shortly summarising the approach by Wong et al. or include this in the methods part as it is important for the reader to understand the difference between the method by Wong et al. and the one presented here.*

A short summary on the model by Wong et al. will be included in a revised version.

> *The ordering of Figures should follow the order they are referred to in the text.*

We agree and we will check the ordering when revising the manuscript.

> *p. 4, l.14ff: "We have...", I do not understand this sentence. Please rephrase.*

Will be rephrased.

> *p. 6. l.4f: Assuming that precipitation is in every case heavy tailed seems like a strong assumption. Could this assumption not also lead to overestimation of extremes as seen for the VGLM in Figure 12?*

As the mixture model is only used for the quantile mapping and not in the VGLM the overestimation of extremes by the VGLM can not be attributed to a heavy tailed mixture model. Note that we also allow for an exponential tail in the mixture model and for a gamma-only version. The possibility of the gamma-only version is explained in the model selection part in the appendix. As suggested by Reviewers #1 and #2 this will be explained in the main manuscript in a revised version.

> *p.7 l.28: The author should give more details for the cross-validation setup. I assume it is in time, but I am not sure which periods have been used for calibration/validation.*

The cross-validation setup will be explained in a revised version.

> *p.11 l. 6: I think that Figure 10d) shows that the model is strongly underestimating the occurence of heavy precipitation events by almost 50% in most locations.*

The model strongly underestimates the occurrence of heavy precipitation events in some locations, particularly in Eastern Europe. This is discussed on p.11, l.7-9. Nevertheless, the occurrence of heavy precipitation is not underestimated in most locations (see Fig. 10d). As also shown by the boxplots in Fig. 11d for many regions the median is close to 5% with relatively small variation as indicated by the size of the box.

*p.12 l.21ff: How is the correlogram calculated for the 100 VGLM realizations. Are the realizations first averaged and is the correlation calculated afterwards or the other way around? Please add this also in the text.*

The correlogram is calculated for the 100 realizations and then averaged. We will explain the calculation of the correlogram more detailed in a revised version.

*p.13 l.9ff: The improvement of the "drizzle effect", "location bias" has not been shown in this study but in previous work. The references should be added to avoid misunderstanding.*

The references will be added in a revised version: For the drizzle effect: Maraun, 2016: Bias correcting climate change simulations – a critical review (Curr Clim Change Rep), and for the location bias: Maraun and Widmann, 2015 which is cited in other parts of the manuscript.

---

## Author Response (AR1)

Dear Dr. Samaniego,

please find enclosed a revised version of our manuscript entitled "A Combined Statistical Bias Correction and Stochastic Downscaling Method for Precipitation".

We implemented all comments of the reviewers. In particular, as major change we added a comparison to the quantile mapping bias correction as it is classically applied, i.e., between RCM and station observations (point scale). We also clarified all parts that where not formulated clear enough and added a paragraph to guide users to the discussion section.

Please find below a detailed point-by-point response to all comments of the reviewers, a list of all relevant changes of the manuscript and the manuscript with highlighted changes.

Yours sincerely,

Claudia Volosciuk (on behalf of the co-authors)

**Point-by-point response to reviewer comments**

We would like to thank the reviewers for carefully reading our manuscript and for their constructive comments which will improve our manuscript. We agree with most suggestions and we implemented them in the revised version.

As major change we added a comparison of our results to the quantile mapping (QM) approach applied between RCM and point scale (i.e., station). In particular, the QM between RCM and point scale (i.e., station) has been added to the spatial autocorrelation plot Fig. 16 of the manuscript (now Fig. 17). The spatial autocorrelation of the QM between RCM and point scale is very similar to that of the QM between RCM and grid scale. This confirms that the QM-approach is not capable to model small scale variability, and that a stochastic model is needed to bridge the scale gap. Particularly in summer (JJA) the VGLM improves spatial autocorrelation compared to QM for both RCM to grid and RCM to point scale. The comparison to QM between RCM and point scale is also added to the intercomparison plots of all models Figs. 14&15 (now Figs. 15&16). In these metrics that evaluate the precipitation distribution the QM between RCM and point scale performs well as expected. To show the problems of QM in representing small scale variability that is not explained by the grid box scale the evaluation of predictive power is required which is here shown by spatial autocorrelation. Note that thereby a plotting error in Figs. 14&15 (now Figs. 15&16) has been corrected (RCM and QM were accidentally plotted inverted). Main conclusions of the manuscript are not affected by this.

Please find a detailed response below.

**Response to Reviewer #1**

MAJOR ISSUES

> *The main motivation of the study relates to the deterministic nature of standard MOS techniques such as QM and their inability to accurately reproduce local-scale variance that is not explained by the actual large-scale predictor. The developed approach is designed to improve on this by separating the bias correction from the downscaling step and by introducing stochasticity into the latter. The performance of both steps and of the combined scheme is evaluated and compared to the performance of (1) raw RCM output and (2) bias-corrected RCM output (bias correction at the resolution of the RCM). What is missing, however, is a comparison to a "standard" QM application that directly bias-corrects and downscales from the grid cell to the point scale (i.e., the first step of the approach directly targeting the stations series instead of the EOBS grid cell). In my opinion, only such a comparison can show the advantages of the new approach compared to standard applications. This is essential in the light of its apparent problems (at many stations the performance of the two-step procedure is worse than raw RCM or bias-corrected RCM output). I'd therefore suggest to include a fourth dataset in the evaluation of the combined approach, i.e., QM of step 1 directly applied to the station series. This might require to rethink the choice of performance metrics, as I'd expect neither the CvM score nor the 95% score to reveal the advantages of introducing stochasticity. Only the spatial autocorrelation might show such improvements. I'm aware that the suggested extension will to some extent be*

*covered by upcoming VALUE papers, but given the motivation of the two-step approach this comparison is essential for the present paper in my opinion.*

We agree with the reviewer that comparing our results to the standard quantile mapping approach from RCM to station data would improve our manuscript and we added this comparison in the revised version (see above). As expected the standard QM between RCM and point scale performs well for the CvM-score and in representing the occurrence of heavy precipitation. However, when evaluating spatial autocorrelation QM to grid scale and QM to point scale are very similar. This reveals that at least in summer with many small scale convective events a stochastic model is required to represent the point scale.

MINOR ISSUES (p: page, l: line)

*Temporal scale of the model calibration: Evaluation results are presented for both winter (DJF) and summer (JJA), but the temporal scale of the calibration of the two steps remains unclear to me. Have all mdoels be fitted separately for winter and summer, or for the full year, or for every doy-of-the-year with a moving window? This information is a detail, but should be provided.*

The model has been calibrated for each season separately. This information is now provided in the explanation of the general concept (p. 4, l. 19).

*Extension to further variables: Climate impact studies often require more downscaled variables than precipitation only. Could the presented approach be applied to other variables (e.g., temperature) as well?*

In principle this approach should be applicable to any variable that is gamma distributed. For, e.g., temperature a model based on a normal distribution might be more suitable. Nevertheless, at this time our approach has only been evaluated for precipitation. Transferring it to other variables would require an evaluation for the particular variable. We now discuss this in the conclusion (p. 19, l. 2-4).

*Performance measures and model selection: An overview of the performance measures (CvM, 95% score, spatial autocorrelation) is not provided, and most information has to be taken from the appendix. It would be helpful for the reader to clearly state in Section 2 or 3 which performance measures are applied. Details can still be covered by the appendix. Also, the "model selection" description is a little scattered and hidden. An additional paragraph in the main part of the manuscript clearly outlining the selection strategy (step 1, later on also the best-performing overall approach) would be helpful.*

We now provide an overview of the applied metrics on p. 9, l. 6-21. Model selection procedures are now described in the main manuscript (p. 7, l. 9-30).

*p1 l19: I'd suggest to replace "precipitation data" by "precipitation projections" as this statement primarily concerns future scenario series.*

Done.

*p2 l18: "leading to too smooth variance in space and time" -> I'd suggest to slightly extend this part and to describe the problems of inflation a little more detailed. Readers not familiar with this issue might have problems to understand this*

*limitation. In my opinion, the variance in time (if measured by the SD) should be rather well captured by QM through inflation.*

We are happy to extend the discussion of this issue. Anyway, also the temporal variability is in general affected by inflation (see, e.g., von Storch, J Climate, 1999). Local precipitation is a randomly disaggregated grid-box precipitation, or vice versa, grid box precipitation is the area average of sub-grid precipitation. The temporal memory is then higher at the grid-box scale than at the local scale, i.e., grid-box time series are smoother in time than local series. Quantile mapping cannot overcome this mismatch in temporal structure (apart from correcting the drizzle effect). This effect is, of course, less pronounced than the spatial effect as demonstrated in Maraun, J. Climate, 2013. Actually, the results of the COST Action VALUE (currently submitted) indicate this effect. We now extended this part (p. 2, l. 24-28).

*p4 l13: Please check: This 0.44◦ RACMO simulation might not be the one that is used in the VALUE perfect predictor experiment mentioned before (there it is the corresponding 0.11◦ simulation to my knowledge). If so, it would be beneficial to briefly mention this fact as it would prevent an inconsistent comparison with the VALUE results.*

The reviewer is right that the RACMO simulation we used differs in resolution from the one used in VALUE. This difference is now mentioned on p. 5, l. 1-2.

*p4, l27-32: The new method is presented as being versatile and applicable to freerunning GCMs/RCMs. While this is true in general, it does not apply to the grid box selection step that relies on temporal correspondence. This should at least briefly be mentioned.*

The reviewer is right that the grid box selection step must be calibrated with a reanalysis-driven RCM-simulation to ensure temporal correspondance. As the location bias is specific to the employed RCM, and most likely not dependent on the driving model, the selected grid box as calibrated by a reanalysis-driven simulation of the same RCM can be applied to the free-running simulation. However, the impact of the driving model on the grid box that best represents local climate should be checked in future work. We now mention this on p. 5, l. 23-24 and discuss it on p. 16, l. 20-23.

*p6 l26: "bias corrected RCM precipitation" (and further occurrences in this section) -> I'd suggest to more generally speak of "coarse-scale precipitation" as not always biascorrected precipitation is used as predictor (in the evaluation of the second step it is EOBS precipitation, and in the evaluation of the full setup it might also be raw RCM precipitation depending on the selected model).*

We agree that the wording suggested by the reviewer is more consistent. We replaced it as suggested in the revised version.

*p8 l5-18: In short, can the remaining inaccuracies of QM be related to non-stationary correction functions (which would show up in the applied cross validation framework)?*

The remaining inaccuracies of QM can be related to both a time-varying correction function as suggested by the reviewer and the parametric correction function. This is now mentioned on p. 10, l. 25-26.

> *Section 5.2 (Evaluation of the second step): To evaluate the second step, the modelled precipitation CDFs are used for the QQ plots which requires a standardization to a stationary gamma distribution. Why not using the same framework as for the evaluation of the combined approach, i.e., drawing 100 realizations of precipitation series and then computing the respective percentiles? This would be more straightforward. Also, why is the evaluation of the second step not carried out within the cross validation framework but within a scheme where calibration and validation periods are identical?*

For the second step we show standardized QQ-Plots where the effect of the predictor is approximately removed. This allows to evaluate the goodness of fit of the model which is our aim here. Drawing 100 realisations would include both the fitted model and the effect of the predictor. This comment of the reviewer clearly shows that we did not explain this sufficiently in the current version of our manuscript. We now clarified this on p. 11, l. 25-28.

The problems with the chosen model in some locations are already present when repredicting the calibration period where the skill should be higher than in a cross validation where a period is predicted that is not part of the calibration period. This clearly highlights deficiencies in the model for these locations even in this setup where the skill should be higher. This is now mentioned on p. 11, l. 18-19 and discussed on p. 12, l. 23-26.

> *p12 l21-34: It remains unclear how the spatial autocorrelation has been computed. Based on seasonal means? Or separately for each day and then averaged?*

The spatial autocorrelation is computed based on daily values and then averaged. The details of how the spatial autocorrelation was computed are now given in the new overview of the evaluation metrics on p. 9, l. 15-21.

> *p14 l14-16: This statement is a little overconfident given the results previously shown. In many regions VGLM is comparable, sometimes even inferior to raw or bias-corrected RCM (as said above, an extra comparison to QM directly onto the station series would be very beneficial here). Concerning the spatial autocorrelation, at least in DJF I wouldn't speak of an improvement by VGLM.*

This part has been reformulated. Here we rather show a concept than the perfect model for each station which we will highlight in a revised manuscript. Further research on the specific implementation is required for some regions. The reviewer is right that in DJF the noise component is slightly too strong in our model. This could be improved by a multi-site model or by including more physical based predictors (i.e., sea level pressure) which is beyond the scope of our study. In DJF a stochastic model is generally less important as more variability is explained by the grid box than in JJA where precipitation is often caused by small scale convective events. It has been rephreased to "*The stochastic downscaling (second step) improves the estimated occurrence of heavy precipitation in many regions* but introduces biases in continental winter climate. *Furthermore, spatial autocorrelation* in JJA *is improved* […]." (p. 17, l. 33-34).  We agree that the extra comparison of the QM between RCM and point scale is beneficial here. The very similar decay of spatial

autocorrelation of QM to grid scale and QM to point scale highlights that at least in summer a stochastic model is required to represent small scale variability.

> *p14 l16-18: This sentence requires a proper reference (this aspect is not covered by the work present).*

This aspect is a simple consequence of using a regression model instead of inflation – in QM, the inflation is responsible for changes in trends. We now cite Maraun, J Climate, 2013, where this issue is explained (p. 18, l. 2-3).

> *User guidance: Given the remaining apparent problems of the new two-step approach at many sites (inferiority even against raw RCM data), some summarizing guidance would be helpful on when to use the new scheme and when this is considered critical. This guidance should account for the fact, that in a free-running setup which is the setup for climate scenarios, not all parts of the presented evaluation can be carried out beforehand (no temporal correspondence which prevents the use of the same strategy for grid box selection and to some extent also the cross-validation setup with short time slices).*

We added a paragraph to guide users in a revised version (p. 18, l. 30-p. 19 l. 4). In this paragraph we highlight that for some regions a specific implementation different from the one we used is required. We now also highlight throughout the manuscript that this work rather introduces a concept than providing the perfect specific implementation for each site (e.g., p. 16, l. 30-33). Moreover, our evaluation shows that bias correction and downscaling methods must not be capriciously transferred from one region to another. Users need to re-evaluate the method when transferring it to locations with different climatic conditions. This aspect is now discussed on p. 18, l. 30-31.

**Response to Reviewer #2**

Major remarks

> *The selection procedure and evaluation in step 1 is not well described, e.g. on p.7 = Sect. 3.1. Usually, I would expect that only bias-corrected data are used in step 2, which is not the case. The selection procedure is well described in the appendix A2 (especially lines 37-40 on p. 16) but not in the main text. Also I would expect that a bias corrected precipitation map is compared with the uncorrected precipitation data and observations. But suddenly a predictor is mentioned instead of precipitation, and a mixed map is shown only for the station locations. This is rather confusing when first reading the paper.*

We now provide an overview of the applied metrics on p. 9, l. 6-21. Model selection procedures are now described in the main manuscript (p. 7, l. 9-30).

We agree with the reviewer that it would be easier for the reader if the results part is started with a more general representation of our results. Thus, we now start the results part with showing maps of the mean bias of the raw uncorrected RCM, the standard quantile mapping between RCM and station, and our method (p. 10, l. 1-9 and Fig. 3).

> *In step 1, the quantile mapping bias correction improves RCM precipitation in 73 of 86 cases in DJF, but only for 49 of 86 cases in JJA. As quantile mapping can be a rather powerful approach, it seems that the chosen gamma function for the transfer*

*functions fails in a lot of cases especially in JJA. Concluding from this it may be suitable to point to approaches (in the discussion section) where several functions can be used as candidate for the transfer function, such it has been done, e.g. by (Piani, C., G.P. Weedon, M. Best, S.M. Gomes, P. Viterbo, S. Hagemann, and J. O. Haerter, 2010: Statistical bias correction of global simulated daily precipitation and temperature for the application of hydrological models. J. Hydrol., 395, doi:10.1016/j.jhydrol.2010.10.024, 199–215)*

We now mention the suggested paper in the discussion section (p. 17, l. 19-22).

*For step 2, it seems even worse. Here, the combined approach provides the best precipitation estimate only for 25 (45) of 86 stations in DJF (JJA). Thus on a first glance, the application of the chosen VLGM does not appear to be a suitable method for the downscaling that can be recommended. What would happen if you use the quantile mapping to directly bias correct the RCM data to the station observations, i.e. using an approach such as it commonly done in bias correction literature? How this would compare to your results?*

As mentioned above a comparison of our results to the QM applied between RCM and point scale (i.e., station) has been added to the revised version of our manuscript. The spatial autocorrelation of the QM between RCM and point scale is very similar to that of the QM between RCM and grid scale (see Fig. 17 in the new manuscript). This confirms that the QM-approach is not capable to model small scale variability, and that a stochastic model is needed to bridge the scale gap. Yet, the reviewer is right that the specific implementation that we employed here is not suitable for all studied locations. We now highlight throughout the manuscript (e.g., p. 16, l. 30-33) that we rather introduce a concept than the perfect specific model for all locations which should be subject of future studies however.

Minor remarks

*p.4 – line 1 Please provide an explanation for readers that are not familiar with the "five-fold cross validation". Either short in the main text or long in the appendix with a reference to this in the main text.*

The five-fold cross validation is now explained when introducing the general concept on p. 4, l. 17-19.

*p.4 – line 19-20 Although E-OBS is probably not an appropriate reference in some regions it …*

Done.

*p.6 – line 28 The term "logit link function" is not common knowledge. Please explain!*

Done (p. 8, l. 12-13).

*p.7 – line 26 …version since the calibration …*

Done.

*p.8 – line 31 … performance of the VGLM gamma for different climates, …*

Done.

> *p.9 – line 5-7 It is written: To evaluate the goodness of fit we use residual QQ-plots (Fig. 6 for DJF and Fig. 8 for JJA). As a QQ-plot requires quantiles of an unconditional distribution we standardized the from day-to-day varying distribution to a stationary gamma distribution (Coles, 2001; Wong et al., 2014). Thereby the effect of the predictor is approximately removed.*
> *What do you mean with "the effect of the predictor is approximately removed"? You are using E-OBS as predictor. If the effect of E-OBS is removed, would you get the same results with any other predictor? I don't understand.*

The parameters of the gamma distribution for a particular day are determined by the estimated regression parameters of the VGLM and the predictor (see Eq. 6 in the manuscript). As the gamma-parameters depend on the predictor they vary from day-to-day due to the varying predictor. The applied standardization transforms these from day-to-day varying gamma-parameters to a stationary gamma distribution. This stationary distribution has no longer the predictor dependent day-to-day variations. Or in other words the effect of the predictor is approximately removed. Due to this procedure the goodness of fit of the regression model can be evaluated instead of evaluating the combined effect of predictor and regression model which is usually present in the from day-to-day varying gamma-parameters. These results should be very similar with any other predictor. Therefore, deficiencies that are indicated by these results are either due to inappropriate model structure or not well fitting parameters which could be caused by an inappropriate calibration dataset.

This comment of the reviewer shows that we did not explain this procedure sufficiently to make it easily understandable for the reader. We now explain this more detailed on p. 11, l. 24-28.

> *p.10 – line 10 …correction,* section *5.1) …*

Done.

> *p.11 – line 29 This* raises *the question …*

Done.

> *p.12 – line 30-32 It is written: " The spatial …improved by the stochastic downscaling step." Obviously this statement is correct for DJF. If half of the regions get worse with the downscaling, this questions the general usefulness of the chosen VGLM method (see also major remarks). In DJF, precipitation is generally strongly determined by the large-scale circulation. Here, the QM bias correction already yields quite good corrected precipitation values. But why the VGLM makes it worse in the majority of cases? This implies a strong weakness of the chosen downscaling method. You provide some potential reasons, but I suggest also coming up with some more details on how this may be improved. You may even undermine this with examples for single stations. For example, you mention "For instance, another distribution in the VGLM…." on p. 15 – line 3-5. Is it possible to provide a plot for one station where another distribution/function is used that improves the downscaling for this particular station?*

The reviewer is right that further research is required to find the appropriate stochastic model for each of the locations. However, this is beyond the scope of our study. As the applied correction function appears to be not flexible enough in some regions we have tried to implement a VGLM with splines. This implementation is unfortunately not straightforward and comes along with the risk of overfitting. We agree that this issue should be implemented and carefully evaluated in a follow-up study. We now highlight throughout the manuscript that the aim of this study is to introduce the concept of combining a bias correction with a downscaling method rather than finding the perfect specific implementation for each location (e.g., 16, l. 30-33). We also discuss now the problems of the VGLM more detailed with pointing to the risk of overfitting in more complex models (p. 12,l. 14-19).

> *Figure 2 I suggest including the PRUDENCE regions in the plot as a major part of your evaluation is based on these regions. For example increase the size of the figure and include PRUDENCE regions as boxes with another colour, e.g. red.*

This suggestion of the reviewer is a very good idea. The PRUDENCE-regions are now shown in Fig. 2 as dashed lines in grey and red for the Alps as this region would otherwise not be distinguishable from the other regions.

> *Figure 3 caption … QM corrected RCM, triangles: uncorrected RCM.*

This is now changed for Fig. 4 (before Fig. 3) and also for Fig. 11 (before Fig. 10).

> *Figure 7 and 9 What benefit do Fig. 7 and 9 provide? Are they necessary or can they be removed?*

The circles in these figures show which predictands there are on the point scale for a given grid scale predictor. The lines show different quantiles of the modeled distribution for the given predictor. On the one hand this shows how well the model fits the observational relationship. While this information is partly redundant to the QQ-plots these plots also provide evidence for potential explanations why the relationship is in some locations not well. The VGLM currently allows basically for three different model behaviors: concave (i.e., Brocken DJF), straight (i.e., San Sebastian DJF) or convex (i.e., Malaga DJF). This appears to be not flexible enough for some locations. No changes between these three types are possible. A more flexible relationship that allows for a changed model behavior for higher values could improve the results but comes along with the risk of overfitting.

Now we provide a more detailed discussion on these figures (p. 12, l. 14-19).

**Response to Reviewer #3**

Major Comments

> *The presented method of VGLM + QM corrected RCM is not able to outperform the low resolution data (i.e. the raw RCM data or the QM corrected data) everywhere. See for example Figure 10, where the RCM (triangle) and QM (circles) is present for most locations. How can it be that a statistical post-processing is decreasing the performance? It should at least be as good as the gridded data. This indicates to me that precipitation is to a large extend not following a gamma distribution as used in the VGLM and that the linear predictor-predictand relationship (eq. 6) is implausible. The authors should provide an explanation on this point and also*

> *relate their findings to that of previous studies such at Wong et al. 2014 or Eden et al. 2014.*

We discuss potential reasons for the problems with the VGLM including an implausible linear relationship in the manuscript. As the reviewer we were also concerned that precipitation might not be gamma-distributed in these regins. Therefore, we verified that precipitation is gamma distributed in these locations. This comment of the reviewer shows that although not shown in the manuscript this shall be mentioned to avoid confusion. We now mention this on p. 12, l. 5.

The studies by Wong et al. and Eden et al. cover the British Isles, and thus, do not tackle the climates where we find problems, i.e., continental winter climates. This shows that bias correction and downscaling methods must not be transferred from one region to another without reevaluation for the climatic conditions of the region where it is transfered to. We now highlight this issue (p. 18, l. 30-32). Now we also mention the agreement for the British Isles (p. 18, l. 11-12).

> *Additionally, there are some contradictions that must be resolved. For example, the authors state on p. 14 l. 26ff, that E-OBS might be unreliable in France and eastern Europe due to low station density that implies a misrepresentation of gridded precipitation in this region. But on p. 12 l. 4, that in Scandinavia E-OBS has a high station density and is of good quality, but the VGLM is still performing poorly. This indicates to me that the quality of E-OBS cannot be identified as the source of bad performance for the VGLM.*

We agree that the quality of E-OBS can not be identified as the only source for bad performance. Nevertheless, we think that it is a potential reason amongst others that could be adressed and quantified in a future study. We clarified this on p. 18, l. 20-21.

> *The results for the different European regions (e.g., Figure 5) should also be related to previous research as tremendous research has used this classification.*

The results for the different European regions are now compared to Dosio and Paruolo, 2011 (JGR). In agreement with our results they get large improvements over the Alps, Spain and France. Yet, in contrast to our results they also get good results for middle and eastern Europe where we find persisting biases even after bias correction (p. 17, l. 17-19).

Minor comment

> *The authors should include an appendix shortly summarising the approach by Wong et al. or include this in the methods part as it is important for the reader to understand the difference between the method by Wong et al. and the one presented here.*

A short summary on the model by Wong et al. is now given in the section "General concept" on p. 3, l. 27-30. The concept of the Wong et al.-model is now also shown in the schematic of the model (Fig. 1).

> *The ordering of Figures should follow the order they are referred to in the text.*

Done.

> *p. 4, l.14ff: "We have...", I do not understand this sentence. Please rephrase.*

Done.

> *p. 6. l.4f: Assuming that precipitation is in every case heavy tailed seems like a strong assumption. Could this assumption not also lead to overestimation of extremes as seen for the VGLM in Figure 12?*

As the mixture model is only used for the quantile mapping and not in the VGLM the overestimation of extremes by the VGLM can not be attributed to a heavy tailed mixture model. Note that we also allow for an exponential tail in the mixture model and for a gamma-only version. The possibility of the gamma-only version is explained in the model selection part which has been moved from the appendix to the main text and is now explained on p. 7. l. 9-18.

> *p.7 l.28: The author should give more details for the cross-validation setup. I assume it is in time, but I am not sure which periods have been used for calibration/validation.*

The cross-validation setup is now explained on p. 4, l. 17-19.

> *p.11 l. 6: I think that Figure 10d) shows that the model is strongly underestimating the occurence of heavy precipitation events by almost 50% in most locations.*

The model strongly underestimates the occurrence of heavy precipitation events in some locations, particularly in Eastern Europe. This is discussed on p.11, l.7-9 in the old manuscript (now: p. 14, l. 6-8). Nevertheless, the occurrence of heavy precipitation is not underestimated in most locations (see Fig. 10d). As also shown by the boxplots in Fig. 11d for many regions the median is close to 5% with relatively small variation as indicated by the size of the box.

> *p.12 l.21ff: How is the correlogram calculated for the 100 VGLM realizations. Are the realizations first averaged and is the correlation calculated afterwards or the other way around? Please add this also in the text.*

The correlogram is calculated for the 100 realizations and then averaged. The estimation of the correlogram is now explained in the new overview of evaluation metrics (p. 9, l. 15-21).

> *p.13 l.9ff: The improvement of the "drizzle effect", "location bias" has not been shown in this study but in previous work. The references should be added to avoid misunderstanding.*

We added the following references: For the drizzle effect: Maraun, 2016: Bias correcting climate change simulations – a critical review (Curr Clim Change Rep), and for the location bias: Maraun and Widmann, 2015 which is cited in other parts of the manuscript.

**List of relevant changes**

- Comparison to classical quantile mapping between RCM and point scale added

- We now highlight throughout the manuscript that our aim is rather to present a proof of concept than the optimal model for each studied location

- Clarification of all unclear issues mentiond by the reviewers

- In General Concept: short description of Wong et al.-model, details of cross-validation and model fitting period added

- Subregions for evaluation added to Fig. 2

- We now mention that the grid-box selection step must be carried out with reanaysis-driven simulation

- Model selection procedure explanation moved to main manuscript

- Evaluation metrics overview added

- Evaluation of mean precipitation bias added

- Explanations clarified and discussion added for evaluation of step 2

- Paragraph for user guidance added in conclusion

[revised manuscript text omitted]

---

## Author Response (AR2)

We thank the editor for his suggestions on our manuscript. We implemented all suggestions. Please find a detailed response below.

*- Center the legend in Fig. 3, put on the right.*
done

*- Fig 4 move legend to the right.*
done

*- Fig 5 is too small. Make a single legend for all panels. Add labels (a)... and refer in caption.*
done, now Fig. 15 in appendix B1

*- Fig 6. Increase separation between panels. Difficult to read. Too small for me.*
done, now Fig. 5

*- Fig 7, 8 Add Labels (a)..., make a single legend.*
done, now Fig. 6 and 16 in appendix B2

*- Fig. 9,10 Add Labels (a)..., make a single legend. Increase separations. Indicate the quantiles on red lines 0.1...(at least in one.)*
done, now Fig. 7 and 17 in appendix B2.

*- Fig. 11 too small. legends to the right. it is not usual to have a legend with integers 0, 1, 2, ..., 32 and reals with three decimals! Please change both.*
done, now Fig. 8. Please note that the number with three decimals is the threshold for the 95%-significance level of the CvM-criterion. Therefore, we left the three decimals. We reduced it to one significance level und we now mention this in the caption.

*- Fig 12 too small, increase separation between panels. Vertical axis has no legend.*
done, now Fig. 9

*- Fig 13,14 same remarks as in Fig 7*
done, now Fig. 10 and 18 in appendix B3.

*- Fig 15,16 Font too small*
Font size increased, now Fig. 11,12

*- Fig 17 dots are too small . make single legend. add labels (a)... include description in caption.*
done, now Fig. 13

*- Fig 18 far too small. Can't read.*
Figure increased

*Fonts of all figures are too small in general. I consider that this manuscript has far too many figures. Many are quite similar. I suggest to condense and select the most important ones and put the others in an appendix.*
Figures 5, 8, 10 and 14  were moved to an appendix as suggested by the editor.

*Regarding the method:*

*In Thober et al (WRR, 2014) 10.1002/2014WR015930, we present a method to downscale RCM/NWP precipitation from monthly to daily values with a temporal cascade and ensuring the crosscorrelation in space. Bárdossy and Pegram, 2009 suggested that copula based methods can be used for the spatial downscaling. The mix of both may the optimal for a spatio-temporal downscaling of precipitation. None of these works have been even mentioned, and many of the references there in. I wonder how thoroughly has been done the literature review in this study. I suggest to put these works in contrast to the approach presented.*

We now mention these referentes where we discuss a possible extension of our single-site approach to a multi-site approach on p. 18, lines 8-10.

[revised manuscript text omitted]